# Differentially Private Reinforcement Learning with Self-Play

**Dan Qiao**
Department of Computer Science & Engineering
University of California, San Diego
San Diego, CA 92093
d2qiao@ucsd.edu

**Yu-Xiang Wang**
Halıcıoğlu Data Science Institute
University of California, San Diego
San Diego, CA 92093
yuxiangw@ucsd.edu

## Abstract

We study the problem of multi-agent reinforcement learning (multi-agent RL) with differential privacy (DP) constraints. This is well-motivated by various real-world applications involving sensitive data, where it is critical to protect users' private information. We first extend the definitions of Joint DP (JDP) and Local DP (LDP) to two-player zero-sum episodic Markov Games, where both definitions ensure trajectory-wise privacy protection. Then we design a provably efficient algorithm based on optimistic Nash value iteration and privatization of Bernstein-type bonuses. The algorithm is able to satisfy JDP and LDP requirements when instantiated with appropriate privacy mechanisms. Furthermore, for both notions of DP, our regret bound generalizes the best known result under the single-agent RL case, while our regret could also reduce to the best known result for multi-agent RL without privacy constraints. To the best of our knowledge, these are the first results towards understanding trajectory-wise privacy protection in multi-agent RL.

## 1 Introduction

This paper considers the problem of multi-agent reinforcement learning (multi-agent RL), wherein several agents simultaneously make decisions in an unfamiliar environment with the goal of maximizing their individual cumulative rewards. Multi-agent RL has been deployed not only in large-scale strategy games like Go [Silver et al., 2017], Poker [Brown and Sandholm, 2019] and MOBA games [Ye et al., 2020], but also in various real-world applications such as autonomous driving [Shalev-Shwartz et al., 2016], negotiation [Bachrach et al., 2020], and trading in financial markets [Shavandi and Khedmati, 2022]. In these applications, the learning agent analyzes users' private feedback in order to refine its performance, where the data from users usually contain sensitive information. Take autonomous driving as an instance, here a trajectory describes the interaction between the cars in a neighborhood during a fixed time window. At each timestamp, given the current situation of each car, the system (central agent) will send a command for each car to take (*e.g.* speed up, pull over), and finally the system gathers the feedback from each car (*e.g.* whether the driving is safe, whether the customer feels comfortable) and enhances its policy. Here, (situation, command, feedback) corresponds to (state, action, reward) in a Markov Game where the state and reward of each user are considered as sensitive information. Therefore, leakage of such information is not acceptable. Regrettably, it has been demonstrated that without the implementation of privacy safeguards, learning agents tend to inadvertently memorize details from individual training data points [Carlini et al., 2019], regardless of their relevance to the learning process [Brown et al., 2021]. This susceptibility exposes multi-agent RL agents to potential privacy threats.

To handle the above privacy issue, Differential privacy (DP) [Dwork et al., 2006] has been widely considered. The output of a differentially private reinforcement learning algorithm cannot be discerned

| Algorithms for Markov Games | Regret without privacy | Regret under $\epsilon$-JDP | Regret under $\epsilon$-LDP |
|---|---|---|---|
| DP-Nash-VI (Our Algorithm 1) | $O(\sqrt{H^2SABT})$ | $O(\sqrt{H^2SABT}+H^3S^2AB/\epsilon)$ | $O(\sqrt{H^2SABT}+S^2AB\sqrt{H^5T}/\epsilon)$ |
| Nash VI [Liu et al., 2021] | $\widetilde{O}(\sqrt{H^2SABT})^*$ | N.A. | N.A. |
| Lower bounds | $\Omega(\sqrt{H^2S(A+B)T})$ [Bai and Jin, 2020] | $\widetilde{\Omega}\left(\sqrt{H^2S(A+B)T}+\frac{HS(A+B)}{\epsilon}\right)$ | $\widetilde{\Omega}\left(\sqrt{H^2S(A+B)T}+\frac{\sqrt{HS(A+B)T}}{\epsilon}\right)$ |
| Algorithms for MDPs ($B=1$) | Regret without privacy | Regret under $\epsilon$-JDP | Regret under $\epsilon$-LDP |
| PUCB [Vietri et al., 2020] | $\widetilde{O}(\sqrt{H^3S^2AT})$ | $\widetilde{O}(\sqrt{H^3S^2AT}+H^3S^2A/\epsilon)^\star$ | N.A. |
| LDP-OBI [Garcelon et al., 2021] | $\widetilde{O}(\sqrt{H^3S^2AT})$ | N.A. | $\widetilde{O}(\sqrt{H^3S^2AT}+S^2A\sqrt{H^5T}/\epsilon)^\dagger$ |
| Private-UCB-VI [Chowdhury and Zhou, 2022] | $\widetilde{O}(\sqrt{H^3SAT})$ | $\widetilde{O}(\sqrt{H^3SAT}+H^3S^2A/\epsilon)$ | $\widetilde{O}(\sqrt{H^3SAT}+S^2A\sqrt{H^5T}/\epsilon)$ |
| DP-UCBVI $^\ddagger$ [Qiao and Wang, 2023] | $\widetilde{O}(\sqrt{H^2SAT})$ | $\widetilde{O}(\sqrt{H^2SAT}+H^3S^2A/\epsilon)$ | $\widetilde{O}(\sqrt{H^2SAT}+S^2A\sqrt{H^5T}/\epsilon)$ |

Table 1: Comparison of our results (in blue) to existing work regarding regret without privacy (*i.e.* the privacy budget is infinity), regret under $\epsilon$-Joint DP and regret under $\epsilon$-Local DP. In the above, $S$ is the number of states, $A, B$ are the number of actions for both players, $H$ is the planning horizon and $K$ is the number of episodes ($T = HK$ is the number of steps). Markov decision processes (MDPs) is a special case of Markov Games where $B = 1$. $*$: This result is the best known regret bound when there is no privacy concern. $\star$: More discussions about this bound can be found in Chowdhury and Zhou [2022]. $\dagger$: The original regret bound in Garcelon et al. [2021] is derived under the setting of stationary MDP, and can be directly transferred to the bound here by adding $\sqrt{H}$ to the first term. $\ddagger$: This algorithm achieved the best known results under single-agent MDPs, and our Algorithm 1 can obtain the same regret bounds under this setting.

from its output in an alternative reality where any specific user is substituted, which effectively mitigates the privacy risks mentioned earlier. However, it is shown [Shariff and Sheffet, 2018] that standard DP will lead to linear regret even under contextual bandits. Therefore, Vietri et al. [2020] considered a relaxed surrogate of DP: *Joint Differential Privacy* (JDP) [Kearns et al., 2014] for RL. Briefly speaking, JDP protects the information about any specific user even given the output of all other users. Meanwhile, another variant of DP: *Local Differential Privacy* (LDP) [Duchi et al., 2013] has also been extended to RL by Garcelon et al. [2021] due to its stronger privacy protection. LDP requires that the raw data of each user is privatized before being sent to the agent. Although following works [Chowdhury and Zhou, 2022, Qiao and Wang, 2023] established near optimal results under these two notions of DP, all of the previous works focused on the single-agent RL setting while the solution to multi-agent RL with differential privacy is still unknown. Therefore we question:

**Question 1.1.** *Is it possible to design a provably efficient self-play algorithm to solve Markov games while satisfying the constraints of differential privacy?*

**Our contributions.** In this paper, we answer the above question affirmatively by proposing a general algorithm for DP multi-agent RL: DP-Nash-VI (Algorithm 1). Our contributions are threefold.

- We first extend the definitions of Joint DP (Definition 2.2) and Local DP (Definition 2.3) to the multi-agent RL setting. Both notions of DP focus on protecting the sensitive information of each trajectory, which is consistent with the counterparts under single-agent RL.

- We design a new algorithm DP-Nash-VI (Algorithm 1) based on optimistic Nash value iteration and privatization of Bernstein-type bonuses. The algorithm can be combined with any Privatizer (for JDP or LDP) that possesses a corresponding regret bound (Theorem 4.1). Moreover, when there is no privacy constraint (*i.e.* the privacy budget is infinity), our regret reduces to the best known regret for non-private multi-agent RL.

- Under the constraint of $\epsilon$-JDP, DP-Nash-VI achieves a regret of $\widetilde{O}(\sqrt{H^2SABT} + H^3S^2AB/\epsilon)$ (Theorem 5.2). Compared to the regret lower bound (Theorem 5.3), the main term is nearly optimal while the additional cost due to JDP has optimal dependence on $\epsilon$. Under the $\epsilon$-LDP constraint, DP-Nash-VI achieves a regret of $\widetilde{O}(\sqrt{H^2SABT} + S^2AB\sqrt{H^5T}/\epsilon)$ (Theorem 5.5), where the dependence on $K, \epsilon$ is optimal according to the lower bound (Theorem 5.6). The pair of results strictly generalizes the best known results for single-agent RL with DP [Qiao and Wang, 2023].

## 1.1 Related work

We compare our results with existing works on differentially private reinforcement learning [Vietri et al., 2020, Garcelon et al., 2021, Chowdhury and Zhou, 2022, Qiao and Wang, 2023] and regret minimization under Markov Games [Liu et al., 2021] in Table 1, while more discussions about differentially private learning algorithms are deferred to Appendix A. Notably, all existing DP RL

algorithms focus on the single-agent case. In comparison, our algorithm works for the more general two-player setting and our results directly match the best known regret bounds [Qiao and Wang, 2023] when applied to the single-agent setting.

Recently, several works provide non-asymptotic theoretical guarantees for learning Markov Games. Bai and Jin [2020] developed the first provably-efficient algorithms in MGs based on optimistic value iteration, and the result is improved by Liu et al. [2021] using model-based approach. Meanwhile, model-free approaches are shown to break the curse of multiagency and improve the dependence on action space [Bai et al., 2020, Jin et al., 2021, Mao et al., 2022, Wang et al., 2023, Cui et al., 2023]. However, all these algorithms base on the original data from users, and thus are vulnerable to various privacy attacks. While several works [Hossain and Lee, 2023, Hossain et al., 2023, Zhao et al., 2023b, Gohari et al., 2023] study the privatization of communications between multiple agents, none of them provide regret guarantees. In comparison, we design algorithms that provably protect the sensitive information in each trajectory, while achieving near-optimal regret bounds simultaneously.

Technically speaking, we follow the idea of optimistic Nash value iteration and privatization of Bernstein-type bonuses. Optimistic Nash value iteration aims to construct both upper bounds and lower bounds for value functions, which could guide the exploration. Such idea has been applied by previous model-based approaches [Bai and Jin, 2020, Liu et al., 2021] to derive tight regret bounds. To satisfy the privacy guarantees, we are required to construct the UCB and LCB privately. In this work, we privatize the transition kernel estimate and construct a private bonus function for our purpose. Among different bonuses, we generalize the approach in Qiao and Wang [2023] and directly operate on the Bernstein-type bonus, which could enable tight regret analysis while the privatization is more technically demanding due to the variance term. To handle this, we first privatize the visitation counts such that they satisfy several nice properties, then we use these counts to construct private transition estimates and private bonuses. Lastly, we manage to prove UCB and LCB, and bound the private terms by their non-private counterparts to complete the regret analysis.

# 2 Problem Setup

We consider reinforcement learning under Markov Games (MGs) [Shapley, 1953] with Differential Privacy (DP) [Dwork et al., 2006]. Below we introduce MGs and define DP under multi-agent RL.

## 2.1 Markov Games and Regret

Markov Games (MGs) are the generalization of Markov Decision Processes (MDPs) to the multi-player setting, where each player aims to maximize her own reward. We consider *two-player zero-sum* episodic MGs, denoted by a tuple $\mathcal{MG} = (\mathcal{S}, \mathcal{A}, \mathcal{B}, H, \{P_h\}_{h=1}^H, \{r_h\}_{h=1}^H, s_1)$, where $\mathcal{S}$ is the state space with $S = |\mathcal{S}|$, $\mathcal{A}$ and $\mathcal{B}$ are the action space for the max-player (who aims to maximize the total reward) and the min-player (who aims to minimize the total reward) respectively with $A = |\mathcal{A}|, B = |\mathcal{B}|$. Besides, $H$ is the horizon while the non-stationary transition kernel $P_h(\cdot|s, a, b)$ gives the distribution of the next state if action $(a, b)$ is taken at state $s$ and time step $h$. In addition, we assume that the reward function $r_h(s, a, b) \in [0, 1]$ is deterministic and known[1]. For simplicity, we assume each episode starts from a fixed initial state $s_1$. Then at each time step $h \in [H]$, two players observe $s_h$ and choose their actions $a_h \in \mathcal{A}$ and $b_h \in \mathcal{B}$ simultaneously, after which both players observe the action of their opponent and receive reward $r_h(s_h, a_h, b_h)$, the environment will transit to $s_{h+1} \sim P_h(\cdot|s_h, a_h, b_h)$.

**Markov policy, value function.** A Markov policy $\mu$ of the max-player can be seen as a series of mappings $\mu = \{\mu_h\}_{h=1}^H$, where each $\mu_h$ maps each state $s \in \mathcal{S}$ to a probability distribution over actions $\mathcal{A}$, *i.e.* $\mu_h : \mathcal{S} \to \Delta(\mathcal{A})$. A Markov policy $\nu$ for the min-player is defined similarly. Given a pair of policies $(\mu, \nu)$ and time step $h \in [H]$, the value function $V_h^{\mu,\nu}(\cdot)$ is defined as $V_h^{\mu,\nu}(s) = \mathbb{E}_{\mu,\nu}[\sum_{t=h}^H r_t | s_h = s]$ while the Q-value function $Q_h^{\mu,\nu}(\cdot, \cdot, \cdot)$ is defined as $Q_h^{\mu,\nu}(s, a, b) = \mathbb{E}_{\mu,\nu}[\sum_{t=h}^H r_t | s_h, a_h, b_h = s, a, b]$ for all $s, a, b$. According to the definitions, the following Bellman equation holds:

$$Q_h^{\mu,\nu}(s, a, b) = [r_h + P_h V_{h+1}^{\mu,\nu}](s, a, b), \quad V_h^{\mu,\nu}(s) = [\mathbb{E}_{\mu,\nu} Q_h^{\mu,\nu}](s), \quad \forall (h, s, a, b).$$

---

[1]This assumption is wlog since the uncertainty of reward is dominated by that of transition kernel.

**Best responses, Nash equilibrium.** For any policy $\mu$ of the max-player, there exists a best response policy $\nu^{\dagger}(\mu)$ of the min-player such that $V_h^{\mu,\nu^{\dagger}(\mu)}(s) = \inf_{\nu} V_h^{\mu,\nu}(s)$ for all $(s,h)$. For simplicity, we denote $V_h^{\mu,\dagger} := V_h^{\mu,\nu^{\dagger}(\mu)}$. Also, $\mu^{\dagger}(\nu)$ and $V_h^{\dagger,\nu}$ can be defined by symmetry. It is shown [Filar and Vrieze, 2012] that there exists a pair of policies $(\mu^{\star}, \nu^{\star})$ that are best responses against each other, *i.e.*, $V_h^{\mu^{\star},\dagger}(s) = V_h^{\mu^{\star},\nu^{\star}}(s) = V_h^{\dagger,\nu^{\star}}(s)$, $\forall (s,h) \in \mathcal{S} \times [H]$. The pair of policies $(\mu^{\star}, \nu^{\star})$ is called the Nash equilibrium of the Markov game, which further satisfies the following minimax property: for all $(s,h) \in \mathcal{S} \times [H]$, $\sup_{\mu} \inf_{\nu} V_h^{\mu,\nu}(s) = V_h^{\mu^{\star},\nu^{\star}}(s) = \inf_{\nu} \sup_{\mu} V_h^{\mu,\nu}(s)$. The value functions of $(\mu^{\star}, \nu^{\star})$ are called Nash value functions and we denote $V_h^{\star} = V_h^{\mu^{\star},\nu^{\star}}, Q_h^{\star} = Q_h^{\mu^{\star},\nu^{\star}}$ for simplicity. Nash equilibrium means that no player could gain more from updating her own policy.

**Learning objective: regret.** Following previous works [Bai and Jin, 2020, Liu et al., 2021], we aim to minimize the regret, which is defined as below:

$$\text{Regret}(K) = \sum_{k=1}^{K} \left[ V_1^{\dagger,\nu^k}(s_1) - V_1^{\mu^k,\dagger}(s_1) \right],$$

where $K$ is the number of episodes the agent interacts with the environment and $(\mu^k, \nu^k)$ are the policies executed by the agent in the $k$-th episode. Note that any sub-linear regret bound can be transferred to a PAC guarantee according to the standard online-to-batch conversion [Jin et al., 2018].

## 2.2 Differential Privacy in Multi-agent RL

For RL with self-play, each trajectory corresponds to the interaction between a pair of users and the environment. The interaction generally follows the protocol below. At time step $h$ of the $k$-th episode, the users send their state $s_h^k$ to a central agent $\mathcal{M}$, then $\mathcal{M}$ sends back a pair of actions $(a_h^k, b_h^k)$ for the users to take, and finally the users send their reward $r_h^k$ to $\mathcal{M}$. Following previous works [Vietri et al., 2020, Chowdhury and Zhou, 2022, Qiao and Wang, 2023], here we let $\mathcal{U} = (u_1, \cdots, u_K)$ denote the sequence of $K$ unique [2] pairs of users who participate in the above RL protocol. Besides, each pair of users $u_k$ is characterized by the $\{s_h^k, r_h^k\}_{h=1}^{H}$ information they would respond to all $(AB)^H$ [3] possible sequences of actions from the agent. Let $\mathcal{M}(\mathcal{U}) = \{(a_h^k, b_h^k)\}_{h,k=1,1}^{H,K}$ denote the whole sequence of actions suggested by the agent $\mathcal{M}$. Then a direct adaptation of differential privacy [Dwork et al., 2006] is defined below, which says that $\mathcal{M}(\mathcal{U})$ and all other pairs excluding $u_k$ together will not disclose much information about user $u_k$.

**Definition 2.1** (Differential Privacy (DP)). *For any $\epsilon > 0$ and $\delta \in [0,1]$, a mechanism $\mathcal{M} : \mathcal{U} \to (\mathcal{A} \times \mathcal{B})^{KH}$ is $(\epsilon, \delta)$-differentially private if for any possible user sequences $\mathcal{U}$ and $\mathcal{U}'$ that is different on one pair of users and any subset $E$ of $(\mathcal{A} \times \mathcal{B})^{KH}$,*

$$\mathbb{P}[\mathcal{M}(\mathcal{U}) \in E] \le e^{\epsilon} \cdot \mathbb{P}[\mathcal{M}(\mathcal{U}') \in E] + \delta.$$

*If $\delta = 0$, we say that $\mathcal{M}$ is $\epsilon$-differentially private ($\epsilon$-DP).*

Unfortunately, privately recommending actions to the pair of users $u_k$ while protecting their own state and reward information is shown to be impractical even for the single-player setting. Therefore, we consider a relaxed version of DP, known as *Joint Differential Privacy* (JDP) [Kearns et al., 2014]. JDP says that for all pairs of users $u_k$, the recommendation to all other pairs excluding $u_k$ will not disclose the sensitive information about $u_k$. Although being weaker than DP, JDP could still provide meaningful privacy protection by ensuring that even if an adversary can observe the interactions between all other users and the environment, it is statistically hard to reconstruct the interaction between $u_k$ and the environment. JDP is first studied by Vietri et al. [2020] under single-agent reinforcement learning, and we extend the definition to the two-player setting.

**Definition 2.2** (Joint Differential Privacy (JDP)). *For any $\epsilon > 0$, a mechanism $\mathcal{M} : \mathcal{U} \to (\mathcal{A} \times \mathcal{B})^{KH}$ is $\epsilon$-joint differentially private if for any $k \in [K]$, any user sequences $\mathcal{U}$ and $\mathcal{U}'$ that is different on the $k$-th pair of users and any subset $E$ of $(\mathcal{A} \times \mathcal{B})^{(K-1)H}$,*

$$\mathbb{P}[\mathcal{M}_{-k}(\mathcal{U}) \in E] \le e^{\epsilon} \cdot \mathbb{P}[\mathcal{M}_{-k}(\mathcal{U}') \in E],$$

---

[2] Uniqueness is assumed wlog, as for a returning user pair one can group them with their previous occurrences.

[3] At each time step $h \in [H]$, the agent suggests actions to both players, and thus there are $AB$ possibilities for each time step $h$.

*where $\mathcal{M}_{-k}(\mathcal{U}) \in E$ means the sequence of actions sent to all pairs of users excluding $u_k$ belongs to set $E$.*

In the example of autonomous driving, JDP ensures that even if an adversary observes the interactions between cars within all time windows except one, it is hard to know what happens during the specific time window. While providing strong privacy protection, JDP requires the central agent $\mathcal{M}$ to have access to the real trajectories from users. However, in various scenarios the users are not even willing to directly share their data with the agent. To address such circumstances, Duchi et al. [2013] developed a stronger notion of privacy named *Local Differential Privacy* (LDP). Now that when considering LDP, the agent can not observe the state of users, we consider the following protocol specific for LDP: at the beginning of the $k$-th episode, the agent $\mathcal{M}$ first sends a policy pair $\pi_k = (\mu_k, \nu_k)$ to the pair of users $u_k$, after running $\pi_k$ and getting a trajectory $X_k$, $u_k$ privatizes their trajectory to $X_k'$ and sends it back to $\mathcal{M}$. We present the definition of Local DP below, which generalizes the LDP under single-agent reinforcement learning by Garcelon et al. [2021]. Briefly speaking, Local DP ensures that it is impractical for an adversary to reconstruct the whole trajectory of $u_k$ even if observing their whole response.

**Definition 2.3** (Local Differential Privacy (LDP))**.** *For any $\epsilon > 0$, a mechanism $\widetilde{\mathcal{M}}$ is $\epsilon$-local differentially private if for any possible trajectories $X, X'$ and any possible set $E \subseteq \{\widetilde{\mathcal{M}}(X) | X \text{ is any possible trajectory}\}$,*

$$\mathbb{P}[\widetilde{\mathcal{M}}(X) \in E] \leq e^{\epsilon} \cdot \mathbb{P}[\widetilde{\mathcal{M}}(X') \in E].$$

In the example of autonomous driving, LDP ensures that the system can only observe a private version of the interactions between cars instead of the raw data.

**Remark 2.4.** *Note that here our definitions of JDP and LDP both provide trajectory-wise privacy protection, which is consistent with previous works [Chowdhury and Zhou, 2022, Qiao and Wang, 2023]. Moreover, under the special case where the min-player plays a fixed and known deterministic policy (or equivalently, $\mathcal{B}$ only contains a single action and $B = 1$), the Markov Game setting reduces to a single-agent Markov decision process while our JDP and LDP directly matches previous definitions for the MDP setting. Therefore, our setting strictly generalizes previous works and requires novel techniques to handle the min-player.*

**Remark 2.5.** *In the following sections we will show that LDP is consistent with sub-linear regret bounds, while it is known that we can not derive sub-linear regret bounds under the constraint of DP. We remark that there is no contradictory since here the RL protocols for DP and LDP are different. As a result, here a guarantee of LDP does not directly imply a guarantee of DP and the two notions are indeed not directly comparable.*

## 3 Algorithm

In this part, we introduce DP-Nash-VI (Algorithm 1). Note that the algorithm takes Privatizer as an input. We analyze the regret of Algorithm 1 for all Privatizers satisfying the Assumption 3.1 below, which includes the cases where the Privatizer is chosen as Central (for JDP) or Local (for LDP).

We first introduce the definition of visitation counts, where $N_h^k(s, a, b) = \sum_{i=1}^{k-1} \mathbb{1}(s_h^i, a_h^i, b_h^i = s, a, b)$ denotes the visitation count of $(s, a, b)$ at time step $h$ until the beginning of the $k$-th episode. Similarly, we let $N_h^k(s, a, b, s') = \sum_{i=1}^{k-1} \mathbb{1}(s_h^i, a_h^i, b_h^i, s_{h+1}^i = s, a, b, s')$ be the visitation count of $(h, s, a, b, s')$ before the $k$-th episode. In multi-agent RL without privacy constraints, such visitation counts are sufficient for estimating the transition kernel $\{P_h\}_{h=1}^H$ and updating the exploration policy, as in previous model-based approaches [Liu et al., 2021]. However, these counts base on the original trajectories from the users, which could reveal sensitive information. Therefore, with the concern of privacy, we can only incorporate these counts after a privacy-preserving step. In other words, we use a Privatizer to transfer the original counts to the private version $\widetilde{N}_h^k(s, a, b), \widetilde{N}_h^k(s, a, b, s')$. We make the following Assumption 3.1 for Privatizer, which says that the private counts are close to real ones. Privatizers for JDP and LDP that satisfy Assumption 3.1 will be proposed in Section 5.

**Assumption 3.1** (Private counts)**.** *For any privacy budget $\epsilon > 0$ and failure probability $\beta \in [0, 1]$, there exists some $E_{\epsilon,\beta} > 0$ such that with probability at least $1 - \beta/3$, for all $(h, s, a, b, s', k) \in [H] \times \mathcal{S} \times \mathcal{A} \times \mathcal{B} \times \mathcal{S} \times [K]$, the $\widetilde{N}_h^k(s, a, b, s')$ and $\widetilde{N}_h^k(s, a, b)$ from Privatizer satisfies:*

---

**Algorithm 1** Differentially Private Optimistic Nash Value Iteration (DP-Nash-VI)

---
1: **Input**: Number of episodes $K$, privacy budget $\epsilon$, failure probability $\beta$ and a Privatizer (can be either Central or Local).
2: **Initialize**: Private counts $\widetilde{N}_h^1(s,a,b) = \widetilde{N}_h^1(s,a,b,s') = 0$ for all $(h,s,a,b,s')$. Set up the confidence bound $E_{\epsilon,\beta}$ w.r.t the Privatizer, the minimal gap $\Delta = H$ and universal constants $C_1, C_2 > 0$. $\iota = \log(30HSABK/\beta)$.
3: **for** $k = 1, 2, \cdots, K$ **do**
4: $\quad \overline{V}_{H+1}^k(\cdot) = \underline{V}_{H+1}^k(\cdot) = 0$.
5: $\quad$ **for** $h = H, H-1, \cdots, 1$ **do**
6: $\quad\quad$ **for** $(s,a,b) \in \mathcal{S} \times \mathcal{A} \times \mathcal{B}$ **do**
7: $\quad\quad\quad$ Compute private transition kernel $\widetilde{P}_h^k(\cdot|s,a,b)$ as in (1).
8: $\quad\quad\quad$ Compute $\gamma_h^k(s,a,b) = \frac{C_1}{H} \cdot \widetilde{P}_h^k(\overline{V}_{h+1}^k - \underline{V}_{h+1}^k)(s,a,b)$.
9: $\quad\quad\quad$ Compute $\Gamma_h^k(s,a,b) = C_2 \sqrt{\dfrac{\mathrm{Var}_{\widetilde{P}_h^k(\cdot|s,a,b)}\left[\left(\frac{\overline{V}_{h+1}^k + \underline{V}_{h+1}^k}{2}\right)(\cdot)\right] \cdot \iota}{\widetilde{N}_h^k(s,a,b)}} + \dfrac{C_2 HSE_{\epsilon,\beta} \cdot \iota}{\widetilde{N}_h^k(s,a,b)} + \dfrac{C_2 H^2 S \iota}{\widetilde{N}_h^k(s,a,b)}$.
10: $\quad\quad\quad$ UCB $\overline{Q}_h^k(s,a,b) = \min\{\sum_{s'} \widetilde{P}_h^k(s'|s,a,b) \cdot \overline{V}_{h+1}^k(s') + [r_h + \gamma_h^k + \Gamma_h^k](s,a,b), H\}$.
11: $\quad\quad\quad$ LCB $\underline{Q}_h^k(s,a,b) = \max\{\sum_{s'} \widetilde{P}_h^k(s'|s,a,b) \cdot \underline{V}_{h+1}^k(s') + [r_h - \gamma_h^k - \Gamma_h^k](s,a,b), 0\}$.
12: $\quad\quad$ **end for**
13: $\quad\quad$ **for** $s \in \mathcal{S}$ **do**
14: $\quad\quad\quad$ Compute the policy $\pi_h^k(\cdot,\cdot|s) = \mathrm{CCE}(\overline{Q}_h^k(s,\cdot,\cdot), \underline{Q}_h^k(s,\cdot,\cdot))$.
15: $\quad\quad\quad$ Compute the value functions $\overline{V}_h^k(s) = \mathbb{E}_{\pi_h^k}\overline{Q}_h^k(s), \quad \underline{V}_h^k(s) = \mathbb{E}_{\pi_h^k}\underline{Q}_h^k(s)$.
16: $\quad\quad$ **end for**
17: $\quad$ **end for**
18: $\quad$ Deploy policy $\pi^k = (\pi_1^k, \cdots, \pi_H^k)$ and get trajectory $(s_1^k, a_1^k, b_1^k, r_1^k, \cdots, s_{H+1}^k)$.
19: $\quad$ Update the private counts to $\widetilde{N}^{k+1}$ via Privatizer.
20: $\quad$ **if** $(\overline{V}_1^k - \underline{V}_1^k)(s_1) < \Delta$ **then**
21: $\quad\quad$ $\Delta = (\overline{V}_1^k - \underline{V}_1^k)(s_1)$ and $\pi^{\mathrm{out}} = \pi^k = (\pi_1^k, \cdots, \pi_H^k)$.
22: $\quad$ **end if**
23: **end for**
24: **Return**: The marginal policies of $\pi^{\mathrm{out}}$: $(\mu^{\mathrm{out}}, \nu^{\mathrm{out}})$.

---

(1) $|\widetilde{N}_h^k(s,a,b,s') - N_h^k(s,a,b,s')| \leq E_{\epsilon,\beta}$, $|\widetilde{N}_h^k(s,a,b) - N_h^k(s,a,b)| \leq E_{\epsilon,\beta}$. $\widetilde{N}_h^k(s,a,b,s') > 0$.
(2) $\widetilde{N}_h^k(s,a,b) = \sum_{s' \in \mathcal{S}} \widetilde{N}_h^k(s,a,b,s') \geq N_h^k(s,a,b)$.

Given the private counts satisfying Assumption 3.1, the private estimate of transition kernel is defined as below.

$$\widetilde{P}_h^k(s'|s,a,b) = \frac{\widetilde{N}_h^k(s,a,b,s')}{\widetilde{N}_h^k(s,a,b)}, \quad \forall (h,s,a,b,s',k). \tag{1}$$

**Remark 3.2.** *Assumption 3.1 is a generalization of Assumption 3.1 of Qiao and Wang [2023] to the two-player setting. The assumption (2) guarantees that the private transition kernel $\widetilde{P}_h^k(\cdot|s,a,b)$ is a valid probability distribution, which enables our usage of Bernstein-type bonus. Besides, $\widetilde{P}$ is close to the empirical transition kernel based on original visitation counts according to Assumption (1).*

**Algorithmic design.** Following previous non-private approaches [Liu et al., 2021], DP-Nash-VI (Algorithm 1) maintains a pair of value functions $\overline{Q}$ and $\underline{Q}$ which are the upper bound and lower bound of the Q value of the current policy when facing best responses (with high probability). More specifically, we use private visitation counts $\widetilde{N}_h^k$ to construct a private estimate of transition kernel $\widetilde{P}_h^k$ (line 7) and a pair of private bonus $\gamma_h^k$ (line 8) and $\Gamma_h^k$ (line 9). Intuitively, the first term of $\Gamma_h^k$ is derived from Bernstein's inequality while the second term is the additional bonus due to differential privacy. Next we do value iteration with bonuses to construct the UCB function $\overline{Q}_h^k$ (line 10) and the LCB function $\underline{Q}_h^k$ (line 11). The policy $\pi^k$ for the $k$-th episode is calculated using the CCE function (discussed below) and we run $\pi^k$ to collect a trajectory (line 14,18). Finally, the Privatizer transfers the

non-private counts to private ones for the next episode (line 19). The output policy $\pi^{\text{out}}$ is chosen as the policy $\pi^k$ with minimal gap $(\overline{V}_1^k - \underline{V}_1^k)(s_1)$ (line 21). Decomposing the output policy, the output policy $(\mu^{\text{out}}, \nu^{\text{out}})$ for both players are the marginal policies of $\pi^{\text{out}}$, *i.e.* $\mu_h^{\text{out}}(\cdot|s) = \sum_{b \in \mathcal{B}} \pi_h^{\text{out}}(\cdot, b|s)$ and $\nu_h^{\text{out}}(\cdot|s) = \sum_{a \in \mathcal{A}} \pi_h^{\text{out}}(a, \cdot|s)$ for all $(h, s) \in [H] \times \mathcal{S}$.

**Coarse Correlated Equilibrium (CCE).** Intuitively speaking, CCE of a Markov Game is a potentially correlated policy where no player could benefit from unilateral unconditional deviation. As a computationally friendly relaxation of Nash Equilibrium, CCE has been applied by previous works [Xie et al., 2020, Liu et al., 2021] to design efficient algorithms. Formally, for any two functions $\overline{Q}(\cdot, \cdot), \underline{Q}(\cdot, \cdot) : \mathcal{A} \times \mathcal{B} \to [0, H]$, $\text{CCE}(\overline{Q}, \underline{Q})$ returns a policy $\pi \in \Delta(\mathcal{A} \times \mathcal{B})$ such that

$$\mathbb{E}_{(a,b)\sim\pi}\overline{Q}(a,b) \geq \max_{a'} \mathbb{E}_{(a,b)\sim\pi}\overline{Q}(a',b), \quad \mathbb{E}_{(a,b)\sim\pi}\underline{Q}(a,b) \leq \min_{b'} \mathbb{E}_{(a,b)\sim\pi}\underline{Q}(a,b').$$

Since Nash Equilibrium (NE) is a special case of CCE and a NE always exists, a CCE always exists. Moreover, a CCE can be derived in polynomial time via linear programming. Note that the policies given by CCE can be correlated for the two players, therefore deploying such policy requires the cooperation of both players (line 18).

# 4 Main results

We first state the regret analysis of DP-Nash-VI (Algorithm 1) based on Assumption 3.1, which can be combined with any Privatizers. The proof of Theorem 4.1 is sketched in Appendix B with details in the Appendix. Note that $(\mu^k, \nu^k)$ denote the marginal policies of $\pi^k$ for both players.

**Theorem 4.1.** *For any privacy budget $\epsilon > 0$, failure probability $\beta \in [0, 1]$ and any Privatizer satisfying Assumption 3.1, with probability at least $1 - \beta$, the regret of DP-Nash-VI (Algorithm 1) is*

$$\text{Regret}(K) = \sum_{k=1}^{K} \left[ V_1^{\dagger,\nu^k}(s_1) - V_1^{\mu^k,\dagger}(s_1) \right] \leq \widetilde{O}\left( \sqrt{H^2 SABT} + H^2 S^2 ABE_{\epsilon,\beta} \right), \quad (2)$$

*where $K$ is the number of episodes and $T = HK$.*

Under the special case where the privacy budget $\epsilon \to \infty$ (*i.e.* there is no privacy concern), plugging $E_{\epsilon,\beta} = 0$ in Theorem 4.1 will imply a regret bound of $\widetilde{O}(\sqrt{H^2 SABT})$. Such result directly matches the best known result for regret minimization without privacy constraints [Liu et al., 2021] and nearly matches the lower bound of $\Omega(\sqrt{H^2 S(A + B)T})$ [Bai and Jin, 2020]. Furthermore, under the special case of single-agent MDP (where $B = 1$), our result reduces to $\text{Regret}(K) \leq \widetilde{O}(\sqrt{H^2 SAT} + H^2 S^2 AE_{\epsilon,\beta})$. Such result matches the best known result under the same set of conditions (Theorem 4.1 of Qiao and Wang [2023]). Therefore, Theorem 4.1 is a generalization of the best known results under MARL [Liu et al., 2021] and Differentially Private (single-agent) RL [Qiao and Wang, 2023] simultaneously.

**PAC guarantee.** Recall that we output a policy $\pi^{\text{out}}$ whose marginal policies are $(\mu^{\text{out}}, \nu^{\text{out}})$. We highlight that the output policy for each player is a single Markov policy that is convenient to store and deploy. Moreover, as a corollary of the regret bound, we give a PAC bound for the output policy.

**Theorem 4.2.** *For any privacy budget $\epsilon > 0$, failure probability $\beta \in [0, 1]$ and any Privatizer that satisfies Assumption 3.1, if the number of episodes satisfies that $K \geq \widetilde{\Omega}\left( \frac{H^3 SAB}{\alpha^2} + \min\left\{ K' | \frac{H^2 S^2 ABE_{\epsilon,\beta}}{K'} \leq \alpha \right\} \right)$, with probability $1 - \beta$, $(\mu^{\text{out}}, \nu^{\text{out}})$ is $\alpha$-**approximate Nash**, i.e., $V_1^{\dagger,\nu^{\text{out}}}(s_1) - V_1^{\mu^{\text{out}},\dagger}(s_1) \leq \alpha$.*

The proof is deferred to Appendix C.4. Here the second term of the sample complexity bound[4] ensures that the additional cost due to DP is bounded by $O(\alpha)$. The detailed PAC guarantees under the special cases where the Privatizer is either Central or Local will be provided in Section 5.

# 5 Privatizers for JDP and LDP

In this section, we propose Privatizers that provide DP guarantees (JDP or LDP) while satisfying Assumption 3.1. The proofs for this section can be found in Appendix D.

---

[4]The presentation here is because the term $E_{\epsilon,\beta}$ is indeed dependent of the number of episodes $K$.

## 5.1 Central Privatizer for Joint DP

Given the number of episodes $K$, the Central Privatizer applies $K$-bounded Binary Mechanism [Chan et al., 2011] to privatize all the visitation counter streams $N_h^k(s,a,b)$, $N_h^k(s,a,b,s')$, thus protecting the information of all single users. Briefly speaking, Binary mechanism takes a stream of partial sums as input and outputs a surrogate stream satisfying differential privacy, while the error for each item scales only logarithmically on the length of the stream[5]. Here in multi-agent RL, for each $(h,s,a,b)$, the stream $\{N_h^k(s,a,b) = \sum_{i=1}^{k-1} \mathbb{1}(s_h^i, a_h^i, b_h^i = s,a,b)\}_{k \in [K]}$ can be considered as the partial sums of $\{\mathbb{1}(s_h^i, a_h^i, b_h^i = s,a,b)\}$. Therefore, after observing $\mathbb{1}(s_h^k, a_h^k, b_h^k = s,a,b)$ at the end of episode $k$, the Binary Mechanism will output a private version of $\sum_{i=1}^{k} \mathbb{1}(s_h^i, a_h^i, b_h^i = s,a,b)$. However, Binary Mechanism alone does not satisfy (2) of Assumption 3.1, and a post-processing step is required. To sum up, we let the Central Privatizer follow the workflow below:

Given the privacy budget for JDP $\epsilon > 0$,

(1) For all $(h,s,a,b,s')$, we apply Binary Mechanism (Algorithm 2 in Chan et al. [2011]) with input parameter $\epsilon' = \frac{\epsilon}{2H \log K}$ to privatize all the visitation counter streams $\{N_h^k(s,a,b)\}_{k \in [K]}$ and $\{N_h^k(s,a,b,s')\}_{k \in [K]}$. We denote the output of Binary Mechanism by $\widehat{N}_h^k$.

(2) The private counts $\widetilde{N}_h^k$ are derived through Section 5.3 with $E_{\epsilon,\beta} = O(\frac{H}{\epsilon} \log(HSABK/\beta)^2)$.

Our Central Privatizer satisfies the privacy guarantee below.

**Lemma 5.1.** *For any possible $\epsilon, \beta$, the Central Privatizer satisfies $\epsilon$-JDP and Assumption 3.1 with $E_{\epsilon,\beta} = \widetilde{O}(\frac{H}{\epsilon})$.*

Combining Lemma 5.1 with Theorem 4.1 and Theorem 4.2, we have the following regret & PAC guarantee under $\epsilon$-JDP.

**Theorem 5.2** (Results under JDP). *For any possible $\epsilon, \beta$, with probability $1 - \beta$, the regret from running DP-Nash-VI (Algorithm 1) instantiated with Central Privatizer satisfies:*

$$\text{Regret}(K) \leq \widetilde{O}(\sqrt{H^2 SABT} + H^3 S^2 AB/\epsilon). \tag{3}$$

*Moreover, if the number of episodes $K$ is larger than $\widetilde{\Omega}(\frac{H^3 SAB}{\alpha^2} + \frac{H^3 S^2 AB}{\epsilon\alpha})$, with probability $1 - \beta$, the output policy $(\mu^{\text{out}}, \nu^{\text{out}})$ is $\alpha$-approximate Nash.*

Similar to the single-agent (MDP) setting ($B = 1$), the additional cost due to JDP is a lower order term under the most prevalent regime where the privacy budget $\epsilon$ is a constant. When applied to the single-agent case, our regret matches the best known regret $\widetilde{O}(\sqrt{H^2 SAT} + H^3 S^2 A/\epsilon)$ [Qiao and Wang, 2023]. Moreover, when compared to the regret lower bound below, our main term is nearly optimal while the lower order term has optimal dependence on $\epsilon$.

**Theorem 5.3.** *For any algorithm Alg satisfying $\epsilon$-JDP, there exists a Markov Game such that the expected regret from running Alg for $K$ episodes ($T = HK$ steps) satisfies:*

$$\mathbb{E}[\text{Regret}(K)] \geq \widetilde{\Omega}(\sqrt{H^2 S(A+B)T} + \frac{HS(A+B)}{\epsilon}).$$

The regret lower bound results from the lower bound for the non-private learning [Bai and Jin, 2020] and an adaptation of the lower bound under JDP guarantees [Vietri et al., 2020] to the multi-player setting. Details are deferred to the appendix.

## 5.2 Local Privatizer for Local DP

At the end of episode $k$, the Local Privatizer perturbs the statistics calculated from the new trajectory before sending it to the agent. Since the set of original visitation counts $\{\sigma_h^k(s,a,b) = \mathbb{1}(s_h^k, a_h^k, b_h^k = s,a,b)\}_{(h,s,a,b)}$ has $\ell_1$ sensitivity $H$, we can achieve $\frac{\epsilon}{2}$-LDP by directly adding Laplace noise, *i.e.*, $\widetilde{\sigma}_h^k(s,a,b) = \sigma_h^k(s,a,b) + \text{Lap}(\frac{2H}{\epsilon})$. Similarly, repeating the above perturbation to $\{\mathbb{1}(s_h^k, a_h^k, b_h^k, s_{h+1}^k = s,a,b,s')\}_{(h,s,a,b,s')}$ will lead to identical results. Therefore, the Local Privatizer with budget $\epsilon$ is as below:

---

[5]More details in Chan et al. [2011] and Kairouz et al. [2021].

(1) We perturb $\sigma_h^k(s, a, b) = \mathbb{1}(s_h^k, a_h^k, b_h^k = s, a, b)$ and $\sigma_h^k(s, a, b, s') = \mathbb{1}(s_h^k, a_h^k, b_h^k, s_{h+1}^k = s, a, b, s')$ by adding independent Laplace noises: for all $(h, s, a, b, s', k)$,

$$\widetilde{\sigma}_h^k(s, a, b) = \sigma_h^k(s, a, b) + \mathrm{Lap}\left(\frac{2H}{\epsilon}\right), \quad \widetilde{\sigma}_h^k(s, a, b, s') = \sigma_h^k(s, a, b, s') + \mathrm{Lap}\left(\frac{2H}{\epsilon}\right). \quad (4)$$

(2) Then the noisy counts are derived according to

$$\widehat{N}_h^k(s, a, b) = \sum_{i=1}^{k-1} \widetilde{\sigma}_h^i(s, a, b), \quad \widehat{N}_h^k(s, a, b, s') = \sum_{i=1}^{k-1} \widetilde{\sigma}_h^i(s, a, b, s'), \quad (5)$$

and the private counts $\widetilde{N}_h^k$ are solved through Section 5.3 with $E_{\epsilon,\beta} = O(\frac{H}{\epsilon}\sqrt{K \log(HSABK/\beta)})$.

Our Local Privatizer satisfies the privacy guarantee below.

**Lemma 5.4.** *For any possible $\epsilon, \beta$, the Local Privatizer satisfies $\epsilon$-LDP and Assumption 3.1 with $E_{\epsilon,\beta} = \widetilde{O}(\frac{H}{\epsilon}\sqrt{K})$.*

Combining Lemma 5.4 with Theorem 4.1 and Theorem 4.2, we have the following regret & PAC guarantee under $\epsilon$-LDP.

**Theorem 5.5** (Results under LDP). *For any possible $\epsilon, \beta$, with probability $1 - \beta$, the regret from running DP-Nash-VI (Algorithm 1) instantiated with Local Privatizer satisfies:*

$$\mathrm{Regret}(K) \leq \widetilde{O}\left(\sqrt{H^2 SABT} + S^2 AB\sqrt{H^5 T}/\epsilon\right). \quad (6)$$

*Moreover, if the number of episodes $K$ is larger than $\widetilde{\Omega}\left(\frac{H^3 SAB}{\alpha^2} + \frac{H^6 S^4 A^2 B^2}{\epsilon^2 \alpha^2}\right)$, with probability $1 - \beta$, the output policy $(\mu^{\mathrm{out}}, \nu^{\mathrm{out}})$ is $\alpha$-approximate Nash.*

Similar to the single-agent case, the additional cost due to LDP is a multiplicative factor to the regret bound. When applied to the single-agent case, our regret matches the best known regret $\widetilde{O}\left(\sqrt{H^2 SAT} + S^2 A\sqrt{H^5 T}/\epsilon\right)$ [Qiao and Wang, 2023]. Moreover, we state the lower bound.

**Theorem 5.6.** *For any algorithm Alg satisfying $\epsilon$-LDP, there exists a Markov Game such that the expected regret from running Alg for $K$ episodes ($T = HK$ steps) satisfies:*

$$\mathbb{E}\left[\mathrm{Regret}(K)\right] \geq \widetilde{\Omega}\left(\sqrt{H^2 S(A+B)T} + \frac{\sqrt{HS(A+B)T}}{\epsilon}\right).$$

The lower bound is adapted from Garcelon et al. [2021]. While our regret has optimal dependence on $\epsilon$ and $K$, the optimal dependence on $H, S, A, B$ remains open.

## 5.3 The post-processing step

Now we introduce the post-processing step. At the end of episode $k$, given the noisy counts $\widehat{N}_h^k(s, a, b)$ and $\widehat{N}_h^k(s, a, b, s')$ for all $(h, s, a, b, s')$, the private visitation counts are constructed as following: for all $(h, s, a, b)$,

$$\left\{\widetilde{N}_h^k(s, a, b, s')\right\}_{s' \in \mathcal{S}} = \operatorname*{argmin}_{\{x_{s'}\}_{s' \in \mathcal{S}}} \max_{s' \in \mathcal{S}} \left|x_{s'} - \widehat{N}_h^k(s, a, b, s')\right|$$

such that $\left|\sum_{s' \in \mathcal{S}} x_{s'} - \widehat{N}_h^k(s, a, b)\right| \leq \frac{E_{\epsilon,\beta}}{4}$ and $x_{s'} \geq 0, \ \forall s'. \quad \widetilde{N}_h^k(s, a, b) = \sum_{s' \in \mathcal{S}} \widetilde{N}_h^k(s, a, b, s').$

$$(7)$$

Lastly, we add a constant term to each count to ensure no underestimation (with high probability).

$$\widetilde{N}_h^k(s, a, b, s') = \widetilde{N}_h^k(s, a, b, s') + \frac{E_{\epsilon,\beta}}{2S}, \quad \widetilde{N}_h^k(s, a, b) = \widetilde{N}_h^k(s, a, b) + \frac{E_{\epsilon,\beta}}{2}. \quad (8)$$

**Remark 5.7.** *Solving problem (7) is equivalent to solving:*

$$\min \ t, \ s.t. \ \left|x_{s'} - \widehat{N}_h^k(s,a,b,s')\right| \le t, \ x_{s'} \ge 0, \ \forall \, s' \in \mathcal{S}, \ \left|\sum_{s' \in \mathcal{S}} x_{s'} - \widehat{N}_h^k(s,a,b)\right| \le \frac{E_{\epsilon,\beta}}{4},$$

*which is a **Linear Programming** problem with $O(S)$ variables and $O(S)$ linear constraints. This can be solved in polynomial time [Nemhauser and Wolsey, 1988]. Note that the computation of CCE (line 14 in Algorithm 1) is also a LP problem, therefore the computational complexity of DP-Nash-VI is dominated by $O(HSABK)$ Linear Programming problems, which is computationally friendly.*

We summarize the properties of private counts $\widetilde{N}_h^k$ below, which says that the post-processing step ensures that our private transition kernel estimate is a valid probability distribution while only enlarging the error by a constant factor.

**Lemma 5.8.** *Suppose $\widehat{N}_h^k$ satisfies that with probability $1 - \frac{\beta}{3}$, uniformly over all $(h,s,a,b,s',k)$,*

$$\left|\widehat{N}_h^k(s,a,b,s') - N_h^k(s,a,b,s')\right| \le \frac{E_{\epsilon,\beta}}{4}, \quad \left|\widehat{N}_h^k(s,a,b) - N_h^k(s,a,b)\right| \le \frac{E_{\epsilon,\beta}}{4},$$

*then the $\widetilde{N}_h^k$ derived above satisfies Assumption 3.1.*

### 5.4 Some discussions

In this part, we generalize the Privatizers in Qiao and Wang [2023] (for single-agent case) to the two-player setting, which enables our usage of Bernstein-type bonuses. Such techniques lead to a tight regret analysis and a near-optimal "non-private part" of the regret bound eventually.

Meanwhile, the additional cost due to DP has sub-optimal dependence on parameters regarding the Markov Game. The issue appears even in the single-agent case and is considered to be inherent to model-based algorithms due to the explicit estimation of private transitions [Garcelon et al., 2021]. The improvement requires new algorithmic designs (*e.g.*, private Q-learning) and we leave those as future works.

Lastly, the Laplace Mechanism can be replaced with other mechanisms, such as Gaussian Mechanism [Dwork et al., 2014] with approximate DP guarantee (or zCDP). The regret and PAC guarantees are readily derived by plugging in the corresponding $E_{\epsilon,\beta}$ to Theorem 4.1 and Theorem 4.2.

## 6 Conclusion

We take the initial steps to study trajectory-wise privacy protection in multi-agent RL. We extend the definitions of Joint DP and Local DP to multi-player RL. In addition, we design a provably-efficient algorithm: DP-Nash-VI (Algorithm 1) that could satisfy either of the two DP constraints with corresponding regret guarantee. Moreover, our regret bounds strictly generalize the best known results under DP single-agent RL. There are various interesting future directions, such as improving the additional cost due to DP via model-free approaches and considering Markov Games with function approximations. We believe the techniques in this paper could serve as basic building blocks.

## Acknowledgments

The research is partially supported by NSF Awards #2007117 and #2048091. The work was done while DQ and YW were with the Department of Computer Science at UCSB.

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

# A   Extended related works

**Differentially private reinforcement learning.** The stream of research on DP RL started from the offline setting. Balle et al. [2016] first studied privately evaluating the value of a fixed policy from running it for several episodes (the on policy setting). Later, Xie et al. [2019] considered a more general setting of DP off policy evaluation. Recently, Qiao and Wang [2022b] provided the first results for offline reinforcement learning with DP guarantees.

More efforts focused on solving regret minimization. Under the setting of tabular MDP, Vietri et al. [2020] designed PUCB by privatizing UBEV [Dann et al., 2017] to satisfy Joint DP. Besides, under the constraints of Local DP, Garcelon et al. [2021] designed LDP-OBI based on UCRL2 [Jaksch et al., 2010]. Chowdhury and Zhou [2022] designed a general framework for both JDP and LDP based on UCBVI [Azar et al., 2017], and improved upon previous results. Finally, the best known results are obtained by Qiao and Wang [2023] via incorporating Bernstein-type bonuses. Meanwhile, Wu et al. [2023b] studied the case with heavy-tailed rewards. Under linear MDP, the only algorithm with JDP guarantee: Private LSVI-UCB [Ngo et al., 2022] is a private and low switching [6] version of LSVI-UCB [Jin et al., 2020], while LDP under linear MDP still remains open. Under linear mixture MDP, LinOpt-VI-Reg [Zhou, 2022] generalized UCRL-VTR [Ayoub et al., 2020] to guarantee JDP, while Liao et al. [2023] also privatized UCRL-VTR for LDP guarantee. In addition, Luyo et al. [2021] provided a unified framework for analyzing joint and local DP exploration.

There are several other works regarding DP RL. Wang and Hegde [2019] proposed privacy-preserving Q-learning to protect the reward information. Ono and Takahashi [2020] studied the problem of distributed reinforcement learning under LDP. Lebensold et al. [2019] presented an actor critic algorithm with differentially private critic. Cundy and Ermon [2020] tackled DP-RL under the policy gradient framework. Chowdhury et al. [2021] considered the adaptive control of differentially private linear quadratic (LQ) systems. Zhao et al. [2023a] studied differentially private temporal difference (TD) learning. Chowdhury et al. [2023] analyzed reward estimation with preference feedback under the constraints of DP. Hossain and Lee [2023], Hossain et al. [2023], Zhao et al. [2023b], Gohari et al. [2023] focused on the privatization of communications between multiple agents in multi-agent RL. For applications, DP RL was applied to protect sensitive information in natural language processing and large language models (LLM) [Ullah et al., 2023, Wu et al., 2023a]. Meanwhile, Zhao et al. [2022] considered linear sketches with DP.

# B   Proof overview

In this section, we provide a proof sketch of Theorem 4.1, which can further imply the PAC guarantee (Theorem 4.2) and the regret bounds under JDP (Theorem 5.2) or LDP (Theorem 5.5). The proof consists of the following steps:

(1) Bound the difference between the private statistics and their non-private counterparts.

(2) Prove that UCB and LCB hold with high probability.

(3) Bound the regret via telescoping over time steps and replace the private terms by non-private ones.

Below we explain the key steps in detail. Recall that $N_h^k$ denotes the real visitation counts, while $\widetilde{N}_h^k, \widetilde{P}_h^k$ are the private visitation counts and private transition kernel respectively.

**Step (1).**   According to Assumption 3.1 and standard concentration inequalities, we provide high probability upper bounds for $\|\widetilde{P}_h^k(\cdot|s,a,b) - P_h(\cdot|s,a,b)\|_1$ and $|\widetilde{P}_h^k(s'|s,a,b) - P_h(s'|s,a,b)|$. Besides, we upper bound the following key term $|(\widetilde{P}_h^k - P_h) \cdot V_{h+1}^\star(s,a,b)|$ by $\widetilde{O}\left(\sqrt{\mathrm{Var}_{\widehat{P}_h^k(\cdot|s,a,b)} V_{h+1}^\star(\cdot)/\widetilde{N}_h^k(s,a,b)} + HSE_{\epsilon,\beta}/\widetilde{N}_h^k(s,a,b)\right)$. Details are deferred to Appendix C.1.

**Step (2).** Then we prove that UCB and LCB hold with high probability via backward induction over timesteps (Appendix C.2). More specifically, the variance term of $\Gamma_h^k$ is the private Bernstein-type

---

[6]For low switching RL, please refer to Qiao et al. [2022], Qiao and Wang [2022a], Qiao et al. [2023], Qiao and Wang [2024].

bonus, while the difference between the private variance and its non-private counterpart can be bounded by $\gamma_h^k$ and the lower order terms in $\Gamma_h^k$.

**Step (3).** Lastly, the regret can be bounded by telescoping:

$$\text{Regret}(K) \leq O\left(\underbrace{\sum_{k=1}^{K} \sum_{h=1}^{H} \Gamma_h^k(s_h^k, a_h^k, b_h^k)}_{\text{bound by non-private terms}}\right)$$

$$\leq \widetilde{O}\left(\underbrace{\sum_{k=1}^{K} \sum_{h=1}^{H} \sqrt{\frac{\text{Var}_{P_h(\cdot|s_h^k,a_h^k,b_h^k)} V_{h+1}^{\pi^k}}{N_h^k(s_h^k, a_h^k, b_h^k)}}}_{\text{bound by Cauchy-Schwarz inequality and L.T.V.}} + \underbrace{\sum_{k=1}^{K} \sum_{h=1}^{H} \frac{HSE_{\epsilon,\beta}}{N_h^k(s_h^k, a_h^k, b_h^k)}}_{\leq H^2 S^2 ABE_{\epsilon,\beta}\iota}\right)$$

$$\leq \widetilde{O}(\sqrt{H^2 SABT} + H^2 S^2 ABE_{\epsilon,\beta}).$$

The details about each inequality above and the lower order terms we ignore are deferred to Appendix C.3.

# C  Proof of main theorems

In this section, we prove Theorem 4.1 and Theorem 4.2.

## C.1  Properties of private estimations

We begin with some concentration results about our private transition kernel estimate $\widetilde{P}$ that will be useful for the proof. Throughout the paper, let the non-private empirical transition kernel be:

$$\widehat{P}_h^k(s'|s, a, b) = \frac{N_h^k(s, a, b, s')}{N_h^k(s, a, b)}, \quad \forall (h, s, a, b, s', k). \tag{9}$$

In addition, recall that our private transition kernel estimate is defined as below.

$$\widetilde{P}_h^k(s'|s, a, b) = \frac{\widetilde{N}_h^k(s, a, b, s')}{\widetilde{N}_h^k(s, a, b)}, \quad \forall (h, s, a, b, s', k). \tag{10}$$

Now we are ready to list the properties below. Note that $\iota = \log(30HSABK/\beta)$ throughout the paper.

**Lemma C.1.** *With probability $1 - \frac{\beta}{15}$, for all $(h, s, a, b, k) \in [H] \times \mathcal{S} \times \mathcal{A} \times \mathcal{B} \times [K]$, it holds that:*

$$\left\|\widetilde{P}_h^k(\cdot|s, a, b) - P_h(\cdot|s, a, b)\right\|_1 \leq 2\sqrt{\frac{S\iota}{\widetilde{N}_h^k(s, a, b)}} + \frac{2SE_{\epsilon,\beta}}{\widetilde{N}_h^k(s, a, b)}, \tag{11}$$

$$\left\|\widetilde{P}_h^k(\cdot|s, a, b) - \widehat{P}_h^k(\cdot|s, a, b)\right\|_1 \leq \frac{2SE_{\epsilon,\beta}}{\widetilde{N}_h^k(s, a, b)}. \tag{12}$$

*Proof of Lemma C.1.* The proof is a direct generalization of Lemma B.2 and Remark B.3 in Qiao and Wang [2023] to the two-player setting. $\square$

**Lemma C.2.** *With probability $1 - \frac{2\beta}{15}$, for all $(h, s, a, b, s', k) \in [H] \times \mathcal{S} \times \mathcal{A} \times \mathcal{B} \times \mathcal{S} \times [K]$, it holds that:*

$$\left|\widetilde{P}_h^k(s'|s, a, b) - P_h(s'|s, a, b)\right| \leq 2\sqrt{\frac{\min\{P_h(s'|s, a, b), \widetilde{P}_h^k(s'|s, a, b)\}\iota}{\widetilde{N}_h^k(s, a, b)}} + \frac{2E_{\epsilon,\beta}\iota}{\widetilde{N}_h^k(s, a, b)}, \tag{13}$$

$$\left|\widetilde{P}_h^k(s'|s, a, b) - \widehat{P}_h^k(s'|s, a, b)\right| \leq \frac{2E_{\epsilon,\beta}}{\widetilde{N}_h^k(s, a, b)}. \tag{14}$$

*Proof of Lemma C.2.* The proof is a direct generalization of Lemma B.4 and Remark B.5 in Qiao and Wang [2023] to the two-player setting. □

**Lemma C.3.** *With probability* $1 - \frac{2\beta}{15}$, *for all* $(h, s, a, b, k) \in [H] \times \mathcal{S} \times \mathcal{A} \times \mathcal{B} \times [K]$, *it holds that:*

$$\left|\left(\widetilde{P}_h^k - P_h\right) \cdot V_{h+1}^\star(s, a, b)\right| \leq \min\left\{\sqrt{\frac{2\mathrm{Var}_{P_h(\cdot|s,a,b)} V_{h+1}^\star(\cdot) \cdot \iota}{\widetilde{N}_h^k(s, a, b)}}, \sqrt{\frac{2\mathrm{Var}_{\widehat{P}_h^k(\cdot|s,a,b)} V_{h+1}^\star(\cdot) \cdot \iota}{\widetilde{N}_h^k(s, a, b)}}\right\} + \frac{2HSE_{\epsilon,\beta}\iota}{\widetilde{N}_h^k(s, a, b)},$$
(15)

$$\left|\left(\widetilde{P}_h^k - \widehat{P}_h^k\right) \cdot V_{h+1}^\star(s, a, b)\right| \leq \frac{2HSE_{\epsilon,\beta}}{\widetilde{N}_h^k(s, a, b)}.$$
(16)

*Proof of Lemma C.3.* The proof is a direct generalization of Lemma B.6 and Remark B.7 in Qiao and Wang [2023] to the two-player setting. □

According to a union bound, the following lemma holds.

**Lemma C.4.** *Under the high probability event that Assumption 3.1 holds, with probability at least* $1 - \frac{\beta}{3}$, *the conclusions in Lemma C.1, Lemma C.2, Lemma C.3 hold simultaneously.*

Throughout the proof, we will assume that Assumption 3.1 and Lemma C.4 hold, which will happen with high probability. Before we prove the main theorems, we present the following lemma which bounds the two variances.

**Lemma C.5** (Lemma C.5 of Qiao and Wang [2022b]). *For any function* $V \in \mathbb{R}^S$ *such that* $\|V\|_\infty \leq H$, *it holds that*

$$\left|\sqrt{\mathrm{Var}_{\widetilde{P}_h^k(\cdot|s,a,b)}(V)} - \sqrt{\mathrm{Var}_{\widehat{P}_h^k(\cdot|s,a,b)}(V)}\right| \leq \sqrt{3}H \cdot \sqrt{\left\|\widetilde{P}_h^k(\cdot|s, a, b) - \widehat{P}_h^k(\cdot|s, a, b)\right\|_1}.$$
(17)

*In addition, according to Lemma C.1, the left hand side can be further bounded by*

$$\left|\sqrt{\mathrm{Var}_{\widetilde{P}_h^k(\cdot|s,a,b)}(V)} - \sqrt{\mathrm{Var}_{\widehat{P}_h^k(\cdot|s,a,b)}(V)}\right| \leq 3H\sqrt{\frac{SE_{\epsilon,\beta}}{\widetilde{N}_h^k(s, a, b)}}.$$
(18)

## C.2 Proof of UCB and LCB

For notational simplicity, for $V \in \mathbb{R}^S$ such that $\|V\|_\infty \leq H$, we define

$$\widetilde{V}_h^k V(s, a, b) = \mathrm{Var}_{\widetilde{P}_h^k(\cdot|s,a,b)} V(\cdot), \quad V_h V(s, a, b) = \mathrm{Var}_{P_h(\cdot|s,a,b)} V(\cdot).$$
(19)

Then the bonus term $\Gamma$ can be represented as below ($C_2$ is the universal constant in Algorithm 1).

$$\Gamma_h^k(s, a, b) = C_2\sqrt{\frac{\widetilde{V}_h^k\left(\frac{\overline{V}_{h+1}^k + \underline{V}_{h+1}^k}{2}\right)(s, a, b) \cdot \iota}{\widetilde{N}_h^k(s, a, b)}} + \frac{C_2 HSE_{\epsilon,\beta} \cdot \iota}{\widetilde{N}_h^k(s, a, b)} + \frac{C_2 H^2 S\iota}{\widetilde{N}_h^k(s, a, b)}.$$
(20)

We state the following lemma that can bound the lower order term, which is helpful for proving UCB and LCB.

**Lemma C.6.** *Suppose Assumption 3.1 and Lemma C.4 hold, then there exists a universal constant* $c_1 > 0$ *such that: if function* $g(s)$ *satisfies* $|g|(s) \leq (\overline{V}_{h+1}^k - \underline{V}_{h+1}^k)(s)$, *then it holds that:*

$$\left|(\widetilde{P}_h^k - P_h)g(s, a, b)\right| \leq \frac{c_1}{H} \min\left\{P_h(\overline{V}_{h+1}^k - \underline{V}_{h+1}^k)(s, a, b), \widetilde{P}_h^k(\overline{V}_{h+1}^k - \underline{V}_{h+1}^k)(s, a, b)\right\}$$
$$+ \frac{c_1 H^2 S\iota}{\widetilde{N}_h^k(s, a, b)} + \frac{c_1 HSE_{\epsilon,\beta}\iota}{\widetilde{N}_h^k(s, a, b)}.$$
(21)

*Proof of Lemma C.6.* If $|g|(s) \leq (\overline{V}_{h+1}^k - \underline{V}_{h+1}^k)(s)$, it holds that:

$$
\left|(\widetilde{P}_h^k - P_h)g(s,a,b)\right| \leq \sum_{s'} \left|\left(\widetilde{P}_h^k - P_h\right)(s'|s,a,b)\right| \cdot |g|(s')
$$

$$
\leq \sum_{s'} \left|\left(\widetilde{P}_h^k - P_h\right)(s'|s,a,b)\right| \cdot \left(\overline{V}_{h+1}^k - \underline{V}_{h+1}^k\right)(s')
$$

$$
\leq \sum_{s'} \left(2\sqrt{\frac{P_h(s'|s,a,b)\iota}{\widetilde{N}_h^k(s,a,b)}} + \frac{2E_{\epsilon,\beta}\iota}{\widetilde{N}_h^k(s,a,b)}\right) \cdot \left(\overline{V}_{h+1}^k - \underline{V}_{h+1}^k\right)(s') \tag{22}
$$

$$
\leq \sum_{s'} \left(\frac{P_h(s'|s,a,b)}{H} + \frac{H\iota}{\widetilde{N}_h^k(s,a,b)} + \frac{2E_{\epsilon,\beta}\iota}{\widetilde{N}_h^k(s,a,b)}\right) \cdot \left(\overline{V}_{h+1}^k - \underline{V}_{h+1}^k\right)(s')
$$

$$
\leq \frac{c_1}{H}P_h(\overline{V}_{h+1}^k - \underline{V}_{h+1}^k)(s,a,b) + \frac{c_1 H^2 S\iota}{\widetilde{N}_h^k(s,a,b)} + \frac{c_1 HSE_{\epsilon,\beta}\iota}{\widetilde{N}_h^k(s,a,b)},
$$

where the third inequality is because of Lemma C.2. The forth inequality results from AM-GM inequality. The last inequality holds for some universal constant $c_1$.

The empirical part with the R.H.S to be $\widetilde{P}_h^k$ can be proven using identical proof according to (13). $\quad\square$

Then we prove that the UCB and LCB functions are actually upper and lower bounds of the best responses. Recall that $\pi^k$ is the (correlated) policy executed in the $k$-th episode and $(\mu^k, \nu^k)$ for both players are the marginal policies of $\pi^k$. In other words, $\mu_h^k(\cdot|s) = \sum_{b\in\mathcal{B}} \pi_h^k(\cdot,b|s)$ and $\nu_h^k(\cdot|s) = \sum_{a\in\mathcal{A}} \pi_h^k(a,\cdot|s)$ for all $(h,s) \in [H] \times \mathcal{S}$.

**Lemma C.7.** *Suppose Assumption 3.1 and Lemma C.4 hold, then there exist universal constants $C_1, C_2 > 0$ (in Algorithm 1) such that for all $(h,s,a,b,k) \in [H] \times \mathcal{S} \times \mathcal{A} \times \mathcal{B} \times [K]$, it holds that:*

$$
\begin{cases}
\overline{Q}_h^k(s,a,b) \geq Q_h^{\dagger,\nu^k}(s,a,b) \geq Q_h^{\mu^k,\dagger}(s,a,b) \geq \underline{Q}_h^k(s,a,b), \\
\overline{V}_h^k(s) \geq V_h^{\dagger,\nu^k}(s) \geq V_h^{\mu^k,\dagger}(s) \geq \underline{V}_h^k(s).
\end{cases} \tag{23}
$$

*Proof of Lemma C.7.* We prove by backward induction. For each $k \in [K]$, the conclusion is obvious for $h = H+1$. Suppose UCB and LCB hold for Q value functions in the $(h+1)$-th time step, we first prove the bounds for V functions in the $(h+1)$-th step and then prove the bounds for Q functions in the $h$-th step. For all $s \in \mathcal{S}$, it holds that

$$
\begin{aligned}
\overline{V}_{h+1}^k(s) &= \mathbb{E}_{\pi_{h+1}^k} \overline{Q}_{h+1}^k(s) \\
&\geq \sup_\mu \mathbb{E}_{\mu,\nu_{h+1}^k} \overline{Q}_{h+1}^k(s) \\
&\geq \sup_\mu \mathbb{E}_{\mu,\nu_{h+1}^k} Q_{h+1}^{\dagger,\nu^k}(s) \\
&= V_{h+1}^{\dagger,\nu^k}(s).
\end{aligned} \tag{24}
$$

The conclusion $\underline{V}_{h+1}^k(s) \leq V_{h+1}^{\mu^k,\dagger}(s)$ can be proven by symmetry. Therefore, it holds that

$$
\overline{V}_{h+1}^k(s) \geq V_{h+1}^{\dagger,\nu^k}(s) \geq V_{h+1}^\star(s) \geq V_{h+1}^{\mu^k,\dagger}(s) \geq \underline{V}_{h+1}^k(s). \tag{25}
$$

Next we prove the bounds for Q value functions at the $h$-th step. For all $(s,a,b)$, it holds that

$$
\left(\overline{Q}_h^k - Q_h^{\dagger,\nu^k}\right)(s,a,b) \geq \min\left\{\left(\widetilde{P}_h^k \overline{V}_{h+1}^k - P_h V_{h+1}^{\dagger,\nu^k} + \gamma_h^k + \Gamma_h^k\right)(s,a,b), 0\right\}
$$

$$
\geq \min\left\{\left(\widetilde{P}_h^k V_{h+1}^{\dagger,\nu^k} - P_h V_{h+1}^{\dagger,\nu^k} + \gamma_h^k + \Gamma_h^k\right)(s,a,b), 0\right\}
$$

$$
= \min\left\{\underbrace{\left(\widetilde{P}_h^k - P_h\right)\left(V_{h+1}^{\dagger,\nu^k} - V_{h+1}^\star\right)(s,a,b)}_{(i)} + \underbrace{\left(\widetilde{P}_h^k - P_h\right)V_{h+1}^\star(s,a,b)}_{(ii)} + \gamma_h^k(s,a,b) + \Gamma_h^k(s,a,b), 0\right\}.
$$

$$
\tag{26}
$$

The absolute value of term (i) can be bounded as below.

$$|(\text{i})| \leq \frac{c_1}{H} \widetilde{P}_h^k (\overline{V}_{h+1}^k - \underline{V}_{h+1}^k)(s,a,b) + \frac{c_1 H^2 S \iota}{\widetilde{N}_h^k(s,a,b)} + \frac{c_1 H S E_{\epsilon,\beta} \iota}{\widetilde{N}_h^k(s,a,b)}, \tag{27}$$

for some universal constant $c_1$ according to Lemma C.6.

The absolute value of term (ii) can be bounded as below.

$$|(\text{ii})| \leq \sqrt{\frac{2 \mathrm{Var}_{\widehat{P}_h^k(\cdot|s,a,b)} V_{h+1}^\star(\cdot) \cdot \iota}{\widetilde{N}_h^k(s,a,b)}} + \frac{2 H S E_{\epsilon,\beta} \iota}{\widetilde{N}_h^k(s,a,b)} \leq \sqrt{\frac{2 \mathrm{Var}_{\widetilde{P}_h^k(\cdot|s,a,b)} V_{h+1}^\star(\cdot) \cdot \iota}{\widetilde{N}_h^k(s,a,b)}} + \frac{8 H S E_{\epsilon,\beta} \iota}{\widetilde{N}_h^k(s,a,b)}, \tag{28}$$

where the first inequality is because of Lemma C.3 while the second inequality holds due to Lemma C.5.

We further bound the term $\mathrm{Var}_{\widetilde{P}_h^k(\cdot|s,a,b)} V_{h+1}^\star(\cdot)$ as below.

$$\left| \widetilde{V}_h^k \left( \frac{\overline{V}_{h+1}^k + \underline{V}_{h+1}^k}{2} \right) - \widetilde{V}_h^k V_{h+1}^\star(\cdot) \right| (s,a,b)$$

$$\leq \left| \widetilde{P}_h^k \cdot \left( \frac{\overline{V}_{h+1}^k + \underline{V}_{h+1}^k}{2} \right)^2 - \widetilde{P}_h^k \cdot \left( V_{h+1}^\star \right)^2 \right| (s,a,b) + \left| \left[ \widetilde{P}_h^k \cdot \left( \frac{\overline{V}_{h+1}^k + \underline{V}_{h+1}^k}{2} \right) (s,a,b) \right]^2 - \left[ \widetilde{P}_h^k V_{h+1}^\star(s,a,b) \right]^2 \right|$$

$$\leq 4 H \widetilde{P}_h^k \cdot \left( \overline{V}_{h+1}^k - \underline{V}_{h+1}^k \right) (s,a,b). \tag{29}$$

Therefore, the term (ii) can be further bounded as below.

$$|(\text{ii})| \leq \sqrt{\frac{2 \mathrm{Var}_{\widetilde{P}_h^k(\cdot|s,a,b)} V_{h+1}^\star(\cdot) \cdot \iota}{\widetilde{N}_h^k(s,a,b)}} + \frac{8 H S E_{\epsilon,\beta} \iota}{\widetilde{N}_h^k(s,a,b)}$$

$$\leq \sqrt{\frac{2 \iota \cdot \widetilde{V}_h^k \left( \frac{\overline{V}_{h+1}^k + \underline{V}_{h+1}^k}{2} \right) (s,a,b) + 2 \iota \cdot 4 H \widetilde{P}_h^k \cdot \left( \overline{V}_{h+1}^k - \underline{V}_{h+1}^k \right) (s,a,b)}{\widetilde{N}_h^k(s,a,b)}} + \frac{8 H S E_{\epsilon,\beta} \iota}{\widetilde{N}_h^k(s,a,b)}$$

$$\leq \sqrt{\frac{2 \widetilde{V}_h^k \left( \frac{\overline{V}_{h+1}^k + \underline{V}_{h+1}^k}{2} \right) (s,a,b) \iota}{\widetilde{N}_h^k(s,a,b)}} + \frac{\widetilde{P}_h^k \cdot \left( \overline{V}_{h+1}^k - \underline{V}_{h+1}^k \right) (s,a,b)}{H} + \frac{2 H^2 \iota}{\widetilde{N}_h^k(s,a,b)} + \frac{8 H S E_{\epsilon,\beta} \iota}{\widetilde{N}_h^k(s,a,b)}, \tag{30}$$

where the second inequality results from (29) and the third inequality is due to AM-GM inequality.

Combining the upper bounds of $|(\text{i})|$ and $|(\text{ii})|$, there exist universal constants $C_1, C_2 > 0$ such that

$$(\text{i}) + (\text{ii}) + \gamma_h^k(s,a,b) + \Gamma_h^k(s,a,b) \geq 0. \tag{31}$$

The inequality implies that $\left( \overline{Q}_h^k - Q_h^{\dagger,\nu^k} \right) (s,a,b) \geq 0$. By symmetry, we have $\left( \underline{Q}_h^k - Q_h^{\mu^k,\dagger} \right) (s,a,b) \leq 0$. As a result, it holds that $\overline{Q}_h^k(s,a,b) \geq Q_h^{\dagger,\nu^k}(s,a,b) \geq Q_h^\star(s,a,b) \geq Q_h^{\mu^k,\dagger}(s,a,b) \geq \underline{Q}_h^k(s,a,b)$.

According to backward induction, the conclusion holds for all $(h,s,a,b,k)$. $\qquad\square$

## C.3 Proof of Theorem 4.1

Given the UCB and LCB property, we are now ready to prove our main results. We first state the following lemma that controls the error of the empirical variance estimator.

**Lemma C.8.** *Suppose Assumption 3.1 and Lemma C.4 hold, then there exists a universal constant $c_2 > 0$ such that for all $(h, s, a, b, k) \in [H] \times \mathcal{S} \times \mathcal{A} \times \mathcal{B} \times [K]$, it holds that*

$$\left| \widetilde{V}_h^k \left( \frac{\overline{V}_{h+1}^k + \underline{V}_{h+1}^k}{2} \right) - V_h V_{h+1}^{\pi^k} \right| (s, a, b)$$

$$\leq 4 H P_h \left( \overline{V}_{h+1}^k - \underline{V}_{h+1}^k \right) (s, a, b) + \frac{c_2 H^2 S E_{\epsilon, \beta}}{\widetilde{N}_h^k(s, a, b)} + c_2 H^2 \sqrt{\frac{S \iota}{\widetilde{N}_h^k(s, a, b)}}. \tag{32}$$

*Proof of Lemma C.8.* According to Lemma C.7, $\overline{V}_h^k(s) \geq V_h^{\pi^k}(s) \geq \underline{V}_h^k(s)$ always holds. Then it holds that

$$\left| \widetilde{V}_h^k \left( \frac{\overline{V}_{h+1}^k + \underline{V}_{h+1}^k}{2} \right) - V_h V_{h+1}^{\pi^k} \right| (s, a, b)$$

$$\leq \left| \widetilde{P}_h^k \left( \frac{\overline{V}_{h+1}^k + \underline{V}_{h+1}^k}{2} \right)^2 - P_h \left( V_{h+1}^{\pi^k} \right)^2 - \left[ \widetilde{P}_h^k \left( \frac{\overline{V}_{h+1}^k + \underline{V}_{h+1}^k}{2} \right) \right]^2 + \left( P_h V_{h+1}^{\pi^k} \right)^2 \right| (s, a, b)$$

$$\leq \left| \widetilde{P}_h^k \left( \overline{V}_{h+1}^k \right)^2 - P_h \left( \underline{V}_{h+1}^k \right)^2 - \left( \widetilde{P}_h^k \underline{V}_{h+1}^k \right)^2 + \left( P_h \overline{V}_{h+1}^k \right)^2 \right| (s, a, b)$$

$$\leq \underbrace{\left| \left( \widetilde{P}_h^k - P_h \right) \left( \overline{V}_{h+1}^k \right)^2 \right| (s, a, b)}_{\text{(i)}} + \underbrace{\left| P_h \left[ \left( \overline{V}_{h+1}^k \right)^2 - \left( \underline{V}_{h+1}^k \right)^2 \right] \right| (s, a, b)}_{\text{(ii)}}$$

$$+ \underbrace{\left| \left( \widetilde{P}_h^k \underline{V}_{h+1}^k \right)^2 - \left( P_h \underline{V}_{h+1}^k \right)^2 \right| (s, a, b)}_{\text{(iii)}} + \underbrace{\left| \left( P_h \underline{V}_{h+1}^k \right)^2 - \left( P_h \overline{V}_{h+1}^k \right)^2 \right| (s, a, b)}_{\text{(iv)}}.$$

$$\tag{33}$$

The term (i) can be bounded as below due to Lemma C.1.

$$\text{(i)} \leq 2 H^2 \sqrt{\frac{S \iota}{\widetilde{N}_h^k(s, a, b)}} + \frac{2 H^2 S E_{\epsilon, \beta}}{\widetilde{N}_h^k(s, a, b)}. \tag{34}$$

The term (ii) can be directly bounded as below.

$$\text{(ii)} \leq 2 H P_h \left( \overline{V}_{h+1}^k - \underline{V}_{h+1}^k \right) (s, a, b). \tag{35}$$

The term (iii) can be bounded as below due to Lemma C.1.

$$\text{(iii)} \leq 2 H \left| \left( \widetilde{P}_h^k - P_h \right) \underline{V}_{h+1}^k \right| (s, a, b) \leq 4 H^2 \sqrt{\frac{S \iota}{\widetilde{N}_h^k(s, a, b)}} + \frac{4 H^2 S E_{\epsilon, \beta}}{\widetilde{N}_h^k(s, a, b)}. \tag{36}$$

The term (iv) can be directly bounded as below.

$$\text{(iv)} \leq 2 H P_h \left( \overline{V}_{h+1}^k - \underline{V}_{h+1}^k \right) (s, a, b). \tag{37}$$

The conclusion holds according the upper bounds of term (i), (ii), (iii) and (iv). $\square$

Finally we prove the regret bound of Algorithm 1.

*Proof of Theorem 4.1.* Our proof base on Assumption 3.1 and Lemma C.4. We define the following notations.

$$\begin{cases} \Delta_h^k = \left( \overline{V}_h^k - \underline{V}_h^k \right) (s_h^k), \\ \zeta_h^k = \Delta_h^k - \left( \overline{Q}_h^k - \underline{Q}_h^k \right) (s_h^k, a_h^k, b_h^k), \\ \xi_h^k = P_h \left( \overline{V}_{h+1}^k - \underline{V}_{h+1}^k \right) (s_h^k, a_h^k, b_h^k) - \Delta_{h+1}^k. \end{cases} \tag{38}$$

Then it holds that $\zeta_h^k$ and $\xi_h^k$ are martingale differences bounded by $H$. In addition, we use the following abbreviations for notational simplicity.

$$\begin{cases} \gamma_h^k = \gamma_h^k(s_h^k, a_h^k, b_h^k), \\ \Gamma_h^k = \Gamma_h^k(s_h^k, a_h^k, b_h^k), \\ N_h^k = N_h^k(s_h^k, a_h^k, b_h^k), \\ \widetilde{N}_h^k = \widetilde{N}_h^k(s_h^k, a_h^k, b_h^k). \end{cases} \tag{39}$$

Then we have the following analysis about $\Delta_h^k$.

$$\Delta_h^k = \zeta_h^k + \left(\overline{Q}_h^k - \underline{Q}_h^k\right)(s_h^k, a_h^k, b_h^k)$$

$$\leq \zeta_h^k + 2\gamma_h^k + 2\Gamma_h^k + \widetilde{P}_h^k\left(\overline{V}_{h+1}^k - \underline{V}_{h+1}^k\right)(s_h^k, a_h^k, b_h^k)$$

$$\leq \zeta_h^k + 2\Gamma_h^k + \left(1 + \frac{2C_1}{H}\right) \cdot \left[\left(1 + \frac{c_1}{H}\right) \cdot P_h\left(\overline{V}_{h+1}^k - \underline{V}_{h+1}^k\right)(s_h^k, a_h^k, b_h^k) + \frac{c_1 H^2 S\iota}{\widetilde{N}_h^k} + \frac{c_1 HSE_{\epsilon,\beta}\iota}{\widetilde{N}_h^k}\right]$$

$$\leq \zeta_h^k + \left(1 + \frac{c_3}{H}\right) \cdot P_h\left(\overline{V}_{h+1}^k - \underline{V}_{h+1}^k\right)(s_h^k, a_h^k, b_h^k) + \frac{c_3 H^2 S\iota}{\widetilde{N}_h^k} + \frac{c_3 HSE_{\epsilon,\beta}\iota}{\widetilde{N}_h^k}$$

$$+ c_3 \underbrace{\sqrt{\frac{\widetilde{V}_h^k\left(\frac{\overline{V}_{h+1}^k + \underline{V}_{h+1}^k}{2}\right)(s_h^k, a_h^k, b_h^k)\iota}{\widetilde{N}_h^k}}}_{(i)},$$

$$\tag{40}$$

where the first inequality holds because of the definition of $\overline{Q}$ and $\underline{Q}$. The second inequality holds due to the definition of $\gamma_h^k$ and Lemma C.6. The last inequality holds for some universal constant $c_3 > 0$.

The term (i) can be further bounded as below according to Lemma C.8 and AM-GM inequality.

$$(i) \leq \sqrt{\frac{V_h V_{h+1}^{\pi^k}(s_h^k, a_h^k, b_h^k)\iota}{\widetilde{N}_h^k}} + \sqrt{\frac{4HP_h\left(\overline{V}_{h+1}^k - \underline{V}_{h+1}^k\right)(s_h^k, a_h^k, b_h^k)\iota}{\widetilde{N}_h^k}} + \frac{H\sqrt{c_2 SE_{\epsilon,\beta}\iota}}{\widetilde{N}_h^k} + c_2\sqrt{\frac{\iota}{\widetilde{N}_h^k}} + \frac{H^2\iota\sqrt{c_2 S}}{\widetilde{N}_h^k}$$

$$\leq \sqrt{\frac{V_h V_{h+1}^{\pi^k}(s_h^k, a_h^k, b_h^k)\iota}{\widetilde{N}_h^k}} + \frac{c_4 P_h\left(\overline{V}_{h+1}^k - \underline{V}_{h+1}^k\right)(s_h^k, a_h^k, b_h^k)}{H} + \frac{c_4 H^2\sqrt{S}\iota}{\widetilde{N}_h^k} + \frac{c_4 H\sqrt{SE_{\epsilon,\beta}\iota}}{\widetilde{N}_h^k} + c_4\sqrt{\frac{\iota}{\widetilde{N}_h^k}},$$

$$\tag{41}$$

where the first inequality results from Lemma C.8 and AM-GM inequality on the last term of (32). The second inequality holds for some universal constant $c_4 > 0$ according to AM-GM inequality.

Plugging in the upper bound of term (i), for some universal constant $c_5 > 0$, it holds that:

$$\Delta_h^k \leq \zeta_h^k + \left(1 + \frac{c_5}{H}\right)\xi_h^k + \left(1 + \frac{c_5}{H}\right)\Delta_{h+1}^k + c_5\sqrt{\frac{V_h V_{h+1}^{\pi^k}(s_h^k, a_h^k, b_h^k)\iota}{\widetilde{N}_h^k}} + c_5\sqrt{\frac{\iota}{\widetilde{N}_h^k}} + \frac{c_5 H^2 S\iota}{\widetilde{N}_h^k} + \frac{c_5 HSE_{\epsilon,\beta}\iota}{\widetilde{N}_h^k}.$$

$$\tag{42}$$

Summing $\Delta_1^k$ over $k \in [K]$, we have for some universal constant $c_6 > 0$, it holds that:

$$\sum_{k=1}^{K}\Delta_1^k \leq \underbrace{\sum_{k=1}^{K}\sum_{h=1}^{H}\left(1 + \frac{c_5}{H}\right)^{h-1}\zeta_h^k}_{(ii)} + \underbrace{\sum_{k=1}^{K}\sum_{h=1}^{H}\left(1 + \frac{c_5}{H}\right)^{h}\xi_h^k}_{(iii)} + c_6\underbrace{\sum_{k=1}^{K}\sum_{h=1}^{H}\sqrt{\frac{V_h V_{h+1}^{\pi^k}(s_h^k, a_h^k, b_h^k)\iota}{\widetilde{N}_h^k}}}_{(iv)}$$

$$+ c_6\underbrace{\sum_{k=1}^{K}\sum_{h=1}^{H}\sqrt{\frac{\iota}{\widetilde{N}_h^k}}}_{(v)} + c_6\underbrace{\sum_{k=1}^{K}\sum_{h=1}^{H}\frac{H^2 S\iota + HSE_{\epsilon,\beta}\iota}{\widetilde{N}_h^k}}_{(vi)}.$$

$$\tag{43}$$

The term (ii) and term (iii) can be bounded by Azuma-Hoeffding inequality. With probability $1 - \frac{2\beta}{9}$, it holds that

$$|(\text{ii})| \leq O\left(\sqrt{H^3 K \iota}\right), \quad |(\text{iii})| \leq O\left(\sqrt{H^3 K \iota}\right). \tag{44}$$

The main term (iv) is bounded as below.

$$
\begin{aligned}
(\text{iv}) &\leq \sum_{k=1}^{K} \sum_{h=1}^{H} \sqrt{\frac{\mathbb{V}_h V_{h+1}^{\pi^k}(s_h^k, a_h^k, b_h^k) \iota}{N_h^k}} \\
&\leq \sqrt{\sum_{k=1}^{K} \sum_{h=1}^{H} \mathbb{V}_h V_{h+1}^{\pi^k}(s_h^k, a_h^k, b_h^k) \iota \cdot \sum_{k=1}^{K} \sum_{h=1}^{H} \frac{1}{N_h^k}} \\
&\leq \sqrt{O\left(H^2 K + H^3 \iota\right) \iota \cdot O(HSAB\iota)} \\
&= \widetilde{O}\left(\sqrt{H^3 SABK} + H^2 \sqrt{SAB}\right).
\end{aligned}
\tag{45}
$$

The first inequality is because $\widetilde{N}_h^k \geq N_h^k$ (Assumption 3.1). The second inequality holds due to Cauchy-Schwarz inequality. The third inequality holds with probability $1 - \frac{\beta}{9}$ because of Law of total variance and standard concentration inequalities (for details please refer to Lemma 8 of Azar et al. [2017]).

The term (v) is bounded as below due to pigeon-hole principle.

$$(\text{v}) \leq \sum_{k=1}^{K} \sum_{h=1}^{H} \sqrt{\frac{\iota}{N_h^k}} \leq O(\sqrt{H^2 SABK\iota}), \tag{46}$$

where the first inequality is because $\widetilde{N}_h^k \geq N_h^k$ (Assumption 3.1). The last one results from pigeon-hole principle.

The term (vi) can be bounded as below.

$$(\text{vi}) \leq \sum_{k=1}^{K} \sum_{h=1}^{H} \frac{H^2 S \iota + HSE_{\epsilon,\beta} \iota}{N_h^k} \leq O(H^3 S^2 AB\iota^2) + O(H^2 S^2 ABE_{\epsilon,\beta}\iota^2). \tag{47}$$

Combining the upper bounds for term $|(\text{ii})|, |(\text{iii})|$, (iv), (v) and (vi). The regret of Algorithm 1 can be bounded as below.

$$
\begin{aligned}
\text{Regret}(K) &= \sum_{k=1}^{K} \left[V_1^{\dagger,\nu^k}(s_1) - V_1^{\mu^k,\dagger}(s_1)\right] \leq \sum_{k=1}^{K} \left[\overline{V}_1^k(s_1) - \underline{V}_1^k(s_1)\right] \\
&= \sum_{k=1}^{K} \Delta_1^k \leq \widetilde{O}\left(\sqrt{H^2 SABT} + H^3 S^2 AB + H^2 S^2 ABE_{\epsilon,\beta}\right),
\end{aligned}
\tag{48}
$$

where $T = HK$ is the number of steps.

The failure probability is bounded by $\beta$ ($\frac{\beta}{3}$ for Assumption 3.1, $\frac{\beta}{3}$ for Lemma C.4, $\frac{\beta}{3}$ for terms (ii), (iii) and (iv)). The proof of Theorem 4.1 is complete. □

### C.4 Proof of Theorem 4.2

In this part, we provide a proof of the PAC guarantee: Theorem 4.2. The proof directly follows from the proof of the regret bound (Theorem 4.1).

*Proof of Theorem 4.2.* Recall that we choose $\pi^{\text{out}} = \pi^{\overline{k}}$ such that $\overline{k} = \operatorname{argmin}_k \left(\overline{V}_1^k - \underline{V}_1^k\right)(s_1)$. Therefore, we have

$$V_1^{\dagger,\nu^{\text{out}}}(s_1) - V_1^{\mu^{\text{out}},\dagger}(s_1) \leq \overline{V}_1^{\overline{k}}(s_1) - \underline{V}_1^{\overline{k}}(s_1) \leq \frac{1}{K} \widetilde{O}\left(\sqrt{H^3 SABK} + H^2 S^2 ABE_{\epsilon,\beta}\right), \tag{49}$$

if ignoring the lower order term of the regret bound.

Therefore, choosing $K \geq \widetilde{\Omega}\left(\frac{H^3 SAB}{\alpha^2} + \min\left\{K' | \frac{H^2 S^2 ABE_{\epsilon,\beta}}{K'} \leq \alpha\right\}\right)$ bounds the R.H.S by $\alpha$. □

# D  Missing proof in Section 5

In this section, we provide the missing proof for results in Section 5. Recall that $N_h^k$ is the real visitation count, $\widehat{N}_h^k$ is the intermediate noisy count calculated by both Privatizers and $\widetilde{N}_h^k$ is the final private count after the post-processing step. Note that most of the proof here are generalizations of Appendix D in Qiao and Wang [2023] to the multi-player setting, and here we state the proof for completeness.

*Proof of Lemma 5.1.* Due to Theorem 3.5 of Chan et al. [2011] and Lemma 34 of Hsu et al. [2014], the release of $\{\widehat{N}_h^k(s,a,b)\}_{(h,s,a,b,k)}$ satisfies $\frac{\epsilon}{2}$-DP. Similarly, the release of $\{\widehat{N}_h^k(s,a,b,s')\}_{(h,s,a,b,s',k)}$ also satisfies $\frac{\epsilon}{2}$-DP. Therefore, the release of the following private counters $\{\widehat{N}_h^k(s,a,b)\}_{(h,s,a,b,k)}$, $\{\widehat{N}_h^k(s,a,b,s')\}_{(h,s,a,b,s',k)}$ satisfy $\epsilon$-DP. Due to post-processing (Lemma 2.3 of Bun and Steinke [2016]), the release of both private counts $\{\widetilde{N}_h^k(s,a,b)\}_{(h,s,a,b,k)}$ and $\{\widetilde{N}_h^k(s,a,b,s')\}_{(h,s,a,b,s',k)}$ also satisfies $\epsilon$-DP. Then it holds that the release of all $\pi^k$ is $\epsilon$-DP according to post-processing. Finally, the guarantee of $\epsilon$-JDP results from Billboard Lemma (Lemma 9 of Hsu et al. [2014]).

For utility analysis, because of Theorem 3.6 of Chan et al. [2011], our choice $\epsilon' = \frac{\epsilon}{2H\log K}$ in Binary Mechanism and a union bound, with probability $1 - \frac{\beta}{3}$, for all $(h,s,a,b,s',k)$,

$$
\begin{aligned}
\left|\widehat{N}_h^k(s,a,b,s') - N_h^k(s,a,b,s')\right| &\leq O\left(\frac{H}{\epsilon}\log(HSABK/\beta)^2\right), \\
\left|\widehat{N}_h^k(s,a,b) - N_h^k(s,a,b)\right| &\leq O\left(\frac{H}{\epsilon}\log(HSABK/\beta)^2\right).
\end{aligned}
\tag{50}
$$

Together with Lemma 5.8, the Central Privatizer satisfies Assumption 3.1 with $E_{\epsilon,\beta} = \widetilde{O}\left(\frac{H}{\epsilon}\right)$. □

*Proof of Theorem 5.2.* The proof directly results from plugging $E_{\epsilon,\beta} = \widetilde{O}\left(\frac{H}{\epsilon}\right)$ into Theorem 4.1 and Theorem 4.2. □

*Proof of Theorem 5.3.* The first term results from the non-private regret lower bound $\Omega(\sqrt{H^2S(A+B)T})$ [Bai and Jin, 2020]. The second term is a direct adaptation of the $\Omega(HSA/\epsilon)$ lower bound for any algorithms with $\epsilon$-JDP guarantee under single-agent MDP [Vietri et al., 2020]. □

*Proof of Lemma 5.4.* The privacy guarantee directly results from properties of Laplace Mechanism and composition of DP [Dwork et al., 2014].

For utility analysis, because of Corollary 12.4 of Dwork et al. [2014] and a union bound, with probability $1 - \frac{\beta}{3}$, for all possible $(h,s,a,b,s',k)$,

$$
\begin{aligned}
\left|\widehat{N}_h^k(s,a,b,s') - N_h^k(s,a,b,s')\right| &\leq O\left(\frac{H}{\epsilon}\sqrt{K\log(HSABK/\beta)}\right), \\
\left|\widehat{N}_h^k(s,a,b) - N_h^k(s,a,b)\right| &\leq O\left(\frac{H}{\epsilon}\sqrt{K\log(HSABK/\beta)}\right).
\end{aligned}
\tag{51}
$$

Together with Lemma 5.8, the Local Privatizer satisfies Assumption 3.1 with $E_{\epsilon,\beta} = \widetilde{O}\left(\frac{H}{\epsilon}\sqrt{K}\right)$. □

*Proof of Theorem 5.5.* The proof directly results from plugging $E_{\epsilon,\beta} = \widetilde{O}\left(\frac{H}{\epsilon}\sqrt{K}\right)$ into Theorem 4.1 and Theorem 4.2. □

*Proof of Theorem 5.6.* The first term results from the non-private regret lower bound $\Omega(\sqrt{H^2S(A+B)T})$ [Bai and Jin, 2020]. The second term is a direct adaptation of the $\Omega(\sqrt{HSAT}/\epsilon)$ lower bound for any algorithms with $\epsilon$-LDP guarantee under single-agent MDP [Garcelon et al., 2021]. □

*Proof of Lemma 5.8.* For clarity, we denote the solution of (7) by $\bar{N}_h^k$ and therefore $\widetilde{N}_h^k(s, a, b, s') = \bar{N}_h^k(s, a, b, s') + \frac{E_{\epsilon,\beta}}{2S}$, $\widetilde{N}_h^k(s, a, b) = \bar{N}_h^k(s, a, b) + \frac{E_{\epsilon,\beta}}{2}$.

When the condition (two inequalities) in Lemma 5.8 holds, the original counts $\{N_h^k(s, a, b, s')\}_{s' \in \mathcal{S}}$ is a feasible solution to the optimization problem, which means that

$$\max_{s'} \left| \bar{N}_h^k(s, a, b, s') - \widehat{N}_h^k(s, a, b, s') \right| \leq \max_{s'} \left| N_h^k(s, a, b, s') - \widehat{N}_h^k(s, a, b, s') \right| \leq \frac{E_{\epsilon,\beta}}{4}.$$

Combining with the condition in Lemma 5.8 with respect to $\widehat{N}_h^k(s, a, b, s')$, it holds that

$$\left| \bar{N}_h^k(s, a, b, s') - N_h^k(s, a, b, s') \right| \leq \left| \bar{N}_h^k(s, a, b, s') - \widehat{N}_h^k(s, a, b, s') \right| + \left| \widehat{N}_h^k(s, a, b, s') - N_h^k(s, a, b, s') \right| \leq \frac{E_{\epsilon,\beta}}{2}.$$

Since $\widetilde{N}_h^k(s, a, b, s') = \bar{N}_h^k(s, a, b, s') + \frac{E_{\epsilon,\beta}}{2S}$ and $\bar{N}_h^k(s, a, b, s') \geq 0$, we have

$$\widetilde{N}_h^k(s, a, b, s') > 0, \quad \left| \widetilde{N}_h^k(s, a, b, s') - N_h^k(s, a, b, s') \right| \leq E_{\epsilon,\beta}. \tag{52}$$

For $\bar{N}_h^k(s, a, b)$, according to the constraints in the optimization problem (7), it holds that

$$\left| \bar{N}_h^k(s, a, b) - \widehat{N}_h^k(s, a, b) \right| \leq \frac{E_{\epsilon,\beta}}{4}.$$

Combining with the condition in Lemma 5.8 with respect to $\widehat{N}_h^k(s, a, b)$, it holds that

$$\left| \bar{N}_h^k(s, a, b) - N_h^k(s, a, b) \right| \leq \left| \bar{N}_h^k(s, a, b) - \widehat{N}_h^k(s, a, b) \right| + \left| \widehat{N}_h^k(s, a, b) - N_h^k(s, a, b) \right| \leq \frac{E_{\epsilon,\beta}}{2}.$$

Since $\widetilde{N}_h^k(s, a, b) = \bar{N}_h^k(s, a, b) + \frac{E_{\epsilon,\beta}}{2}$, we have

$$N_h^k(s, a, b) \leq \widetilde{N}_h^k(s, a, b) \leq N_h^k(s, a, b) + E_{\epsilon,\beta}. \tag{53}$$

According to the last line of the optimization problem (7), we have $\bar{N}_h^k(s, a, b) = \sum_{s' \in \mathcal{S}} \bar{N}_h^k(s, a, b, s')$ and therefore,

$$\widetilde{N}_h^k(s, a, b) = \sum_{s' \in \mathcal{S}} \widetilde{N}_h^k(s, a, b, s'). \tag{54}$$

The proof is complete by combining (52), (53) and (54). $\qquad\square$

