# OpenReview forum: "Differentially Private Reinforcement Learning with Self-Play"
_NeurIPS.cc/2024/Conference — NeurIPS 2024 poster_

### Official Review · Reviewer_f4t2 · 2024-07-08

**Soundness:** 4
**Presentation:** 3
**Contribution:** 3
**Rating:** 7
**Confidence:** 4

**Summary:**

The paper studied two-player zero-sum episodic Markov Games under JDP and LDP. The authors designed DP-Nash-VI algorithm for the problems and derives both upper bounds and lower bounds.

**Strengths:**

1. The paper investigated interesting problem of  two-player zero-sum episodic Markov Games under JDP and LDP.
2. It is good to derive the  the best known regret for non-private multi-agent RL as byproduct.
3. The authors give algorithms design and solid proofs of upper bounds and lower bounds for the problems.

**Weaknesses:**

There is no experimental result to verify their theoretical findings.

**Questions:**

1. In this paper, you consider bounded reward case. Can your method extend to heavy-tailed reward case in [1]?
2. Is it possible to further improve the gaps between your upper bounds and lower bounds?

[1] Yulian Wu, Xingyu Zhou, Sayak Ray Chowdhury, and Di Wang. Differentially private episodic reinforcement learning with heavy-tailed rewards. arXiv preprint arXiv:2306.01121, 2023b

**Limitations:**

There is no experiment but I understand the main contributions of paper is on theoretical side.

---

> ### Author Rebuttal · Authors · 2024-08-06
>
> We appreciate your high quality review and the positive score. Below we will reply to your comments.
>
> **There is no experimental result to verify their theoretical findings.**
>
> Thanks for the comment, we will conduct some experiments in the next version.
>
> **In this paper, you consider bounded reward case. Can your method extend to heavy-tailed reward case in [1]?**
>
> This is a very good question. [1] handled the heavy-tailed reward using a truncation step. Then a privatization step is followed. We are not sure whether this can be extended to the multi-player setting, while we believe incorporating the truncation step to deal with heavy-tailed rewards is an interesting future direction.
>
> **Is it possible to further improve the gaps between your upper bounds and lower bounds?**
>
> The gap for the JDP case is mostly on the lower order term, while there is a gap on the main term under the LDP case. Since our algorithm generalizes the best-known result under the single-agent RL case, we believe the practical way to close the gap is to improve the regret bound under the single-agent RL case. [2] argues that the extra dependency on the parameters may be inherent to model-based algorithms due to the explicit estimation of private rewards and transitions. Therefore, a possible direction is to privatize model-free algorithms, which is still an open problem in the literature.
>
> [2] Evrard Garcelon, Vianney Perchet, Ciara Pike-Burke, and Matteo Pirotta. Local differential privacy for regret minimization in reinforcement learning.
>
> Thanks again for the high-quality review. We hope our response could address your main concerns and we are happy to answer any further questions.

---

> > ### Comment · Reviewer_f4t2 · 2024-08-08
> >
> > Thank you for the reply. I keep the score. Good luck!

---

### Official Review · Reviewer_3xco · 2024-07-11

**Soundness:** 3
**Presentation:** 3
**Contribution:** 3
**Rating:** 5
**Confidence:** 2

**Summary:**

The authors address multi-agent self-play reinforcement learning (RL) with differential privacy (DP) constraints to protect sensitive data. They propose an efficient algorithm that meets JDP and LDP requirements, and its regret bounds generalize the best-known results for single-agent RL, marking the first study of trajectory-wise privacy in multi-agent RL.

**Strengths:**

1. The proposed algorithm demonstrates a statistically tight regret bound, supported by the authors' derived lower bound.
2. The differential privacy analysis is comprehensive, encompassing both Joint Differential Privacy (JDP) and Local Differential Privacy (LDP).
3. The writing is generally clear and concise.

**Weaknesses:**

1. The overall technical contribution seems limited. Could the authors emphasize their primary technical contributions? Specifically, is it feasible to address the problem setting using existing algorithms augmented with the Laplacian mechanism?
2. The absence of an experimental study, even a simple one, is noticeable. Conducting experiments is crucial to validate the efficacy of the proposed algorithm, especially given the authors' claims of its efficiency.

**Questions:**

1. The technical convenience of using Definition 2.2 instead of Definition 2.1 is not clear. Can the authors confirm if their proposed algorithm also satisfies the differential privacy requirements of Definition 2.1? If it does not, could you provide an intuitive explanation?

2. Given that the reward value is known and determined, why does the agent still require reward feedback from the user (as mentioned in line 135)?

---

> ### Author Rebuttal · Authors · 2024-08-06
>
> We appreciate your high quality review and the positive score. Below we will reply to your comments.
>
> **The overall technical contribution seems limited. Could the authors emphasize their primary technical contributions? Specifically, is it feasible to address the problem setting using existing algorithms augmented with the Laplacian mechanism?**
>
> First of all, almost all the algorithms in the DP-RL literature are based on some well-known non-private algorithms. For instance, Private-UCB-VI and DP-UCBVI (in our Table 1) are both based on the famous UCBVI algorithm. Therefore, we choose the non-private algorithm Nash-VI as a base algorithm for designing private self-play algorithms. In addition, we only apply the technique for privatizing visitation counts from [Qiao and Wang, 2023], the construction of private bonus here is different. In the two player setting, we need to handle the other player, and the upper (lower) bound is for the Q value of the current policy when facing best responses. Therefore, the bonus is more complex compared to the single-agent setting. We manage to design a new private bonus term for the two-player setting, prove the validity of optimism and pessimism, and derive a near-optimal regret bound. These are the main techinical contributions of the paper.
>
> **The absence of an experimental study, even a simple one, is noticeable. Conducting experiments is crucial to validate the efficacy of the proposed algorithm, especially given the authors' claims of its efficiency.**
>
> Thanks for the comment, we will conduct some experiments in the next version.
>
> **The technical convenience of using Definition 2.2 instead of Definition 2.1 is not clear. Can the authors confirm if their proposed algorithm also satisfies the differential privacy requirements of Definition 2.1? If it does not, could you provide an intuitive explanation?**
>
> Our algorithm satisfies Def 2.2 but could not satisfy Def 2.1. Indeed, Def 2.1 is not consistent with a sublinear regret bound, even for the simpler single-agent RL setting and contextual bandit setting. An intuitive explanation is that Def 2.1 requires the agent to privately recommend an action to the user while protecting her own state, where a constant regret is inevitable in each episode in the worst case. Therefore, our algorithm with sublinear regret bound could not satisfy Def 2.1.
>
> **Given that the reward value is known and determined, why does the agent still require reward feedback from the user (as mentioned in line 135)?**
>
> The RL protocol we introduced is for the general case where the reward can be stochastic. The DP guarantees are also defined for the general setting. Actually our techniques can be easily extended to handle the setting with stochastic rewards, and the assumption of known rewards is only for the ease of presentation.
>
> Thanks again for the high-quality review. We hope our response could address your main concerns and we are happy to answer any further questions.

---

> > ### Comment · Reviewer_3xco · 2024-08-09
> >
> > Thank you very much for the response. I will keep my score.

---

### Official Review · Reviewer_vYkZ · 2024-07-13

**Soundness:** 4
**Presentation:** 4
**Contribution:** 3
**Rating:** 6
**Confidence:** 3

**Summary:**

This paper explores multi-agent reinforcement learning (RL) with differential privacy (DP) constraints. The authors extend the concepts of Joint DP (JDP) and Local DP (LDP) to two-player zero-sum episodic Markov Games. They develop a provably efficient algorithm that combines optimistic Nash value iteration with the privatization of Bernstein-type bonuses, ensuring satisfaction of JDP and LDP requirements with appropriate privacy mechanisms. The algorithm achieves a regret bound that generalizes the best-known results in single-agent RL and could reduce to the best-known results in multi-agent RL without privacy constraints.

**Strengths:**

1. This paper extends the concepts of Joint DP (JDP) and Local DP (LDP) to two-player zero-sum episodic Markov Games, which is important and inspiring for future studies on differential privacy in multi-agent RL.

2. The proposed DP-Nash-VI algorithm could satisfy either Joint DP (JDP) and Local DP (LDP)constraints with corresponding regret guarantees. Their regret bounds strictly generalize the best-known results under DP single-agent RL, and their results could be reduced to the best-known results in multi-agent RL without privacy constraints.

**Weaknesses:**

Though this is a purely theoretical paper, it may be better to include some simulation results to validate the theoretical results.

**Questions:**

Could the authors highlight the key technical challenges involved in combining the techniques from Liu et al. [2021] and Qiao and Wang [2023]?

**Limitations:**

No negative societal impact of this work.

---

> ### Author Rebuttal · Authors · 2024-08-06
>
> We appreciate your high quality review and the positive score. Below we will reply to your comments.
>
> **Though this is a purely theoretical paper, it may be better to include some simulation results to validate the theoretical results.**
>
> Thanks for the comment, we will conduct some experiments in the next version.
>
> **Could the authors highlight the key technical challenges involved in combining the techniques from Liu et al. [2021] and Qiao and Wang [2023]?**
>
> First of all, almost all the algorithms in the DP-RL literature are based on some well-known non-private algorithms. For instance, Private-UCB-VI and DP-UCBVI (in our Table 1) are both based on the famous UCBVI algorithm. Therefore, we choose the non-private algorithm Nash-VI as a base algorithm for designing private self-play algorithms. In addition, we only apply the technique for privatizing visitation counts from [Qiao and Wang, 2023], the construction of private bonus here is different. In the two player setting, we need to handle the other player, and the upper (lower) bound is for the Q value of the current policy when facing best responses. Therefore, the bonus is more complex compared to the single-agent setting. We manage to design a new private bonus term for the two-player setting, prove the validity of optimism and pessimism, and derive a near-optimal regret bound. These are the main techinical contributions of the paper.
>
> Thanks again for the high-quality review. We hope our response could address your main concerns and we are happy to answer any further questions.

---

> > ### Comment · Reviewer_vYkZ · 2024-08-07
> >
> > Thanks for your reply. I keep my score. Good luck!

---

### Official Review · Reviewer_Bzin · 2024-07-13

**Soundness:** 3
**Presentation:** 3
**Contribution:** 2
**Rating:** 5
**Confidence:** 3

**Summary:**

This paper gives an algorithm for differentially private reinforcement learning in two-player zero-sum games. The paper considers a standard model for differential privacy already established for single-agent RL. In this model, in each episode a unique user follows a policy $\pi$ recommended by the RL agent i.e. the user encounters states $s$, takes actions $a$ sampled from $\pi(s)$, and receives rewards $r$. The goal is for the RL agent to learn optimal policy recommendations, without revealing the private information of each user consisting of the trajectory of states, actions, and rewards. The paper considers two standard models for privacy in RL, joint differential privacy (JDP) and local differential privacy (LDP). The proposed algorithm achieves nearly optimal (in certain parameter regimes) regret in both of these privacy constrained settings.

**Strengths:**

Differential privacy in multi-agent reinforcement learning is an important problem, and achieving this in two-player zero-sum games could be step towards the general multi-agent case.

**Weaknesses:**

- The algorithm proposed is a straightforward combination of prior work [1] achieving privacy for single-agent RL and [2] achieving low-regret learning for self-play in zero-sum games. It is not clear from the paper what new ideas, if any, are needed, beyond directly applying the private counts and bonuses from [1] to compute the upper and lower confidence bounds used in [2].

- While privacy for multi-agent RL seems quite relevant and interesting, privacy for two-player zero-sum games seems much less well-motivated. For example, in the autonomous driving case, general multi-agent RL corresponds to learning in a setting where there are many autonomous vehicles on the road and one wants to keep the information of each one private. Two-player zero-sum markov games instead correspond to the setting where there are exactly two autonomous vehicles on the road in each episode, and somehow they are in direct zero-sum competition (e.g. a one-on-one race). In fact, the only reason differential privacy makes sense in this setting is that the paper assumes that a different pair of users competes in each episode, and it is privacy across these different pairs that is preserved. In general, it really seems to me that the most important questions regarding privacy in RL relate to a large number of interacting agents, and that the setting of this paper was chosen specifically so that the techniques of [1] and [2] could be directly applied, rather than because the problem itself seemed important to solve.

Specific Issues:

- The text in Table 1 is too small to read.

[1] Qiao, Dan, and Yu-Xiang Wang. "Near-optimal differentially private reinforcement learning." International Conference on Artificial Intelligence and Statistics. PMLR, 2023.

[2] Liu, Qinghua, et al. "A sharp analysis of model-based reinforcement learning with self-play." International Conference on Machine Learning. PMLR, 2021.

**Questions:**

1. Are there any technical challenges to overcome when combining the known algorithm for private single-agent RL with the algorithm for self-play in zero-sum games?
2. Why is privacy across episodes of two-player zero-sum games an interesting problem? Is there some natural approach to generalize to mutli-agent RL? Are there natural examples where one would want to preserve privacy in a sequence of direct competitions between two players?

**Limitations:**

Yes.

---

> ### Author Rebuttal · Authors · 2024-08-06
>
> We appreciate your high quality review. Below we will reply to your comments.
>
> **The algorithm proposed is a straightforward combination of prior work [1] achieving privacy for single-agent RL and [2] achieving low-regret learning for self-play in zero-sum games. It is not clear from the paper what new ideas, if any, are needed, beyond directly applying the private counts and bonuses from [1] to compute the upper and lower confidence bounds used in [2].**
>
> First of all, almost all the algorithms in the DP-RL literature are based on some well-known non-private algorithms. For instance, Private-UCB-VI and DP-UCBVI (in our Table 1) are both based on the famous UCBVI algorithm. Therefore, we choose the non-private algorithm Nash-VI as a base algorithm for designing private self-play algorithms. In addition, we only apply the technique for privatizing visitation counts from [Qiao and Wang, 2023], the construction of private bonus here is different. In the two player setting, we need to handle the other player, and the upper (lower) bound is for the Q value of the current policy when facing best responses. Therefore, the bonus is more complex compared to the single-agent setting. We manage to design a new private bonus term for the two-player setting, prove the validity of optimism and pessimism, and derive a near-optimal regret bound. These are the main techinical contributions of the paper.
>
> **While privacy for multi-agent RL seems quite relevant and interesting, privacy for two-player zero-sum games seems much less well-motivated. Is there some natural approach to generalize to mutli-agent RL?**
>
> We agree that privacy for two-player zero-sum games is not as important as the multi-agent case, while we believe the progression of science is based on several small steps. This is the first paper considering trajectory-wise privacy protection in the multi-agent RL setting, and we believe this can be an important middle step towards understanding the role of privacy in multi-agent RL. Regarding extension to the case with more agents, some techniques in this paper can be readily applied. The privatization of visitation counts and the bonus can be combined with current model-based MARL algorithms, and a result like
> Regret $\leq$ non-private regret + addition cost due to DP can be expected. The issue of such approach is that model-based approaches generally suffer from a regret with dependence $\Pi_i A_i$ ($A_i$ is the number of actions for the i-th player). To overcome such issue, we need to privatize model-free algorithms, which is still an open problem in the DP-RL literature, and we leave this as future work.
>
> **The text in Table 1 is too small to read.**
>
> Thanks for the comment, we will edit it to improve readability.
>
> Thanks again for the high-quality review. We hope our response could address your main concerns and we are happy to answer any further questions. We would greatly appreciate it if you could consider raising your score.

---

> > ### Comment · Reviewer_Bzin · 2024-08-13
> >
> > Thanks for your response. I appreciate your points about having to modify the bonus, and about the obstructions regarding the use of model-based algorithms in multi-agent RL. After reviewing the discussion I will increase my score to 5.

---

> > > ### Author Response · Authors · 2024-08-14
> > >
> > > Thanks again for your high-quality review and your support.

---

> ### Comment · Area_Chair_8A1p · 2024-08-13
>
> Dear reviewer,
>
> Since the discussion deadline is approaching, please let the authors know if their rebuttal has addressed your concerns. If you still have concerns, you could first acknowledge that you have read the rebuttal and discuss them in the remaining time (with the authors) or the next phase (with other reviewers).
>
> Best,
>
> Your AC

---

### Decision · Program_Chairs · 2024-09-25

**Decision:**

Accept (poster)

**Comment:**

This paper takes the first step in studying privacy protection in Markov games from a theoretical perspective. It considers both central and local models of DP and establishes corresponding regret bounds that capture the cost due to privacy. The reviewers appreciate the contribution, and all tend to accept it.

One possible improvement the authors could consider is to motivate the problem in more detail, i.e., by highlighting the importance of privacy protection in zero-sum Markov games.